# Benefits of Early Stopping in Gradient Descent
# for Overparameterized Logistic Regression

**Jingfeng Wu** [1]  **Peter L. Bartlett** [* 1 2]  **Matus Telgarsky** [* 3]  **Bin Yu** [* 1]

## Abstract

In overparameterized logistic regression, gradient descent (GD) iterates diverge in norm while converging in direction to the maximum $\ell_2$-margin solution—a phenomenon known as the implicit bias of GD. This work investigates additional regularization effects induced by early stopping in well-specified high-dimensional logistic regression. We first demonstrate that the excess logistic risk vanishes for early-stopped GD but diverges to infinity for GD iterates at convergence. This suggests that early-stopped GD is well-calibrated, whereas asymptotic GD is statistically inconsistent. Second, we show that to attain a small excess zero-one risk, polynomially many samples are sufficient for early-stopped GD, while exponentially many samples are necessary for any interpolating estimator, including asymptotic GD. This separation underscores the statistical benefits of early stopping in the overparameterized regime. Finally, we establish nonasymptotic bounds on the norm and angular differences between early-stopped GD and $\ell_2$-regularized empirical risk minimizer, thereby connecting the implicit regularization of GD with explicit $\ell_2$-regularization.

## 1. Introduction

Modern machine learning often operates in the *overparameterized* regime, where the number of parameters exceeds the number of training data. Despite this, models trained by *gradient descent* (GD) often generalize well even in the absence of explicit regularization (Zhang et al., 2021; Neyshabur et al., 2017; Bartlett et al., 2021). The common

explanation is that GD exhibits certain *implicit regularization* effects that prevent overfitting.

The implicit regularization of GD is relatively well understood in regression settings. Using overparameterized linear regression as an example, amongst all interpolators, GD asymptotically converges to the minimum $\ell_2$-norm interpolator (Zhang et al., 2021). Moreover, when the data covariance satisfies certain conditions, the minimum $\ell_2$-norm interpolator achieves vanishing excess risk while fitting training data with *constant* amount of noise, a phenomenon known as *benign overfitting* (see Bartlett et al., 2020; Tsigler & Bartlett, 2023, and references therein). When the data covariance is general, although benign overfitting may not occur, early-stopped GD (and one-pass stochastic GD) can still achieve vanishing excess risk (Bühlmann & Yu, 2003; Yao et al., 2007; Lin & Rosasco, 2017; Dieuleveut & Bach, 2016; Zou et al., 2023; 2022; Wu et al., 2022a). This suggests early stopping provides an additional regularization effect for GD in linear regression. Moreover, the statistical effects of early stopping are known to be comparable to that of $\ell_2$-regularization in linear regression (Suggala et al., 2018; Ali et al., 2019; Zou et al., 2021; Sonthalia et al., 2024).

However, the picture is less complete for classification, where the risk is measured by the logistic loss and the zero-one loss instead of the squared loss. In overparameterized logistic regression, GD diverges in norm while converging in direction to the maximum $\ell_2$-margin solution (see Soudry et al., 2018; Ji & Telgarsky, 2018, and Proposition 2.2 in Section 2), which is in contrast with GD's convergence to the (bounded!) minimum $\ell_2$-norm solution in the linear regression setting. In standard (finite-dimensional, low-noise, large margin) classification settings, the asymptotic implicit bias of GD implies generalization via classical margin theory (Bartlett & Shawe-Taylor, 1999). More recently, certain high-dimensional settings exhibit well-behaved maximum margin solutions and benign overfitting (see Montanari et al., 2019, for example), but it is unclear if these results apply more broadly or represent special cases. Moreover, if the maximum $\ell_2$-margin solution generalizes poorly, new techniques are required, as the aforementioned least squares techniques cannot be easily adapted owing to their

---

[*]Equal contribution  [1]University of California, Berkeley [2]Google DeepMind [3]New York University. Correspondence to: Jingfeng Wu <uuujf@berkeley.edu>, Peter L. Bartlett <peter@berkeley.edu>, Matus Telgarsky <mjt10041@nyu.edu>, Bin Yu <binyu@berkeley.edu>.

*Proceedings of the 42nd International Conference on Machine Learning*, Vancouver, Canada. PMLR 267, 2025. Copyright 2025 by the author(s).

heavy dependence upon the explicit linear algebraic form of GD's path specific to least squares.

**Contributions.** This work investigates the beneficial regularization effects of early stopping in GD for overparameterized logistic regression. We focus on a well-specified setting where the feature vector follows an anisotropic Gaussian design and the binary label conditional on the feature is given by a logistic model (see Assumption 1 in Section 2). We are particularly interested in the regime where the label contains a constant level of noise. We establish the following results.

1. **Calibration via early stopping.** We first derive risk upper bounds for early-stopped GD that can be applied in the overparameterized regime. With an oracle-chosen stopping time, early-stopped GD achieves vanishing excess logistic risk and excess zero-one error (as the sample size grows) for *every* well-specified logistic regression problem. Furthermore, its naturally induced conditional probability approaches the true underlying conditional probability model. These properties suggest that early-stopped GD is *consistent* and *calibrated* for *every* well-specified logistic regression problem, even in the overparameterized regime.

2. **Advantages over interpolation.** We then provide negative results for GD without early stopping. We show that GD at convergence, in contrast to the typical successes of maximum margin predictors, suffers from an *unbounded* logistic risk and a *constant* calibration error in the overparameterized regime. Moreover, for a broad class of overparameterized logistic regression problems, to attain a small excess zero-one error, early-stopped GD only needs *polynomially* many samples, whereas any interpolating estimators, including asymptotic GD, requires at least *exponentially* many samples. These results underscore the statistical benefits of early stopping.

3. **Connections to $\ell_2$-regularization.** Finally, we compare the GD path (formed by GD iterates with all possible stopping times) with the $\ell_2$-regularization path (formed by $\ell_2$-regularized empirical risk minimizers with all possible regularization strengths). For general convex and smooth problems, including logistic regression, these two paths differ in norm by a factor between $0.585$ and $3.415$, and differ in direction by an angle no more than $\pi/4$. Specific to overparameterized logistic regression, the $\ell_2$-distance of the two paths is asymptotically zero in a widely considered situation but may diverge to infinity in the worst case. These findings partially explain the implicit regularization of early stopping via its connections with the explicit $\ell_2$-regularization.

**Notation.** For two positive-valued functions $f(x)$ and $g(x)$, we write $f(x) \lesssim g(x)$ or $f(x) \gtrsim g(x)$ if there exists

a constant $c > 0$ such that $f(x) \leq cg(x)$ or $f(x) \geq cg(x)$ for every $x$, respectively. We write $f(x) \asymp g(x)$ if $f(x) \lesssim g(x) \lesssim f(x)$. We use the standard big-O notation. For two vectors $\mathbf{u}$ and $\mathbf{v}$ in a Hilbert space, we denote their inner product by $\langle \mathbf{u}, \mathbf{v} \rangle$ or equivalently, $\mathbf{u}^\top \mathbf{v}$. For two matrices $\mathbf{A}$ and $\mathbf{B}$ of appropriate dimension, we define their inner product as $\langle \mathbf{A}, \mathbf{B} \rangle := \text{tr}(\mathbf{A}^\top \mathbf{B})$. For a positive semi-definite (PSD) matrix $\mathbf{A}$ and a vector $\mathbf{v}$ of appropriate dimension, we write $\|\mathbf{v}\|_\mathbf{A}^2 := \mathbf{v}^\top \mathbf{A} \mathbf{v}$. In particular, we write $\|\mathbf{v}\| := \|\mathbf{v}\|_\mathbf{I}$. For a positive integer $n$, we write $[n] := \{1, \dots, n\}$.

## 2. Preliminaries

Let $(\mathbf{x}, y) \in \mathbb{H} \otimes \{\pm 1\}$ be a pair of features and the corresponding binary label sampled from an unknown population distribution. Here $\mathbb{H}$ is a finite or countably infinite dimensional Hilbert space. For a parameter $\mathbf{w} \in \mathbb{H}$, define its population *logistic risk* as

$$\mathcal{L}(\mathbf{w}) := \mathbb{E}\ell(y\mathbf{x}^\top \mathbf{w}), \text{ where } \ell(t) := \ln(1 + e^{-t}),$$

and define its population *zero-one error* as

$$\mathcal{E}(\mathbf{w}) := \mathbb{E}\mathbb{1}\left[y\mathbf{x}^\top \mathbf{w} \leq 0\right] = \Pr(y\mathbf{x}^\top \mathbf{w} \leq 0),$$

where the expectation is over the population distribution of $(\mathbf{x}, y)$. It is worth noting that, different from the logistic risk $\mathcal{L}(\mathbf{w})$, the zero-one error $\mathcal{E}(\mathbf{w})$ is insensitive to the parameter norm. Moreover, we measure the *calibration error* of a parameter $\mathbf{w} \in \mathbb{H}$ by

$$\mathcal{C}(\mathbf{w}) := \mathbb{E}\big|p(\mathbf{w}; \mathbf{x}) - \Pr(y = 1|\mathbf{x})\big|^2,$$

where $p(\mathbf{w}; \mathbf{x})$ is a naturally induced conditional probability given by

$$p(\mathbf{w}; \mathbf{x}) := \frac{1}{1 + \exp(-\mathbf{x}^\top \mathbf{w})}.$$

We say an estimator $\hat{\mathbf{w}}$ is *consistent* (for classification) if it attains the Bayes zero-one error asymptotically, that is, $\mathcal{E}(\hat{\mathbf{w}}) - \min \mathcal{E} \to 0$. We say an estimator $\hat{\mathbf{w}}$ is *calibrated* if its induced conditional probability predicts the true one asymptotically (Foster & Vohra, 1998), that is, $\mathcal{C}(\hat{\mathbf{w}}) \to 0$.

**Gradient descent.** Let $(\mathbf{x}_i, y_i)_{i=1}^n$ be $n$ independent copies of $(\mathbf{x}, y)$. Define the empirical risk as

$$\widehat{\mathcal{L}}(\mathbf{w}) := \frac{1}{n}\sum_{i=1}^n \ell(y_i\mathbf{x}_i^\top \mathbf{w}), \quad \mathbf{w} \in \mathbb{H}.$$

Then the iterates of *gradient descent* (GD) are given by

$$\mathbf{w}_0 = 0, \quad \mathbf{w}_{t+1} = \mathbf{w}_t - \eta\nabla\widehat{\mathcal{L}}(\mathbf{w}_t), \quad t \geq 0, \quad \text{(GD)}$$

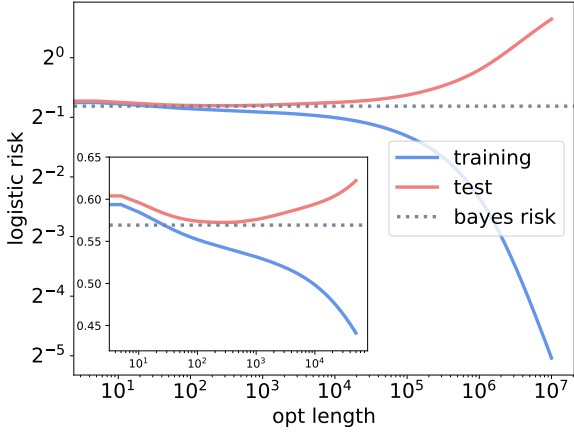

(a) Empirical and population logistic risk

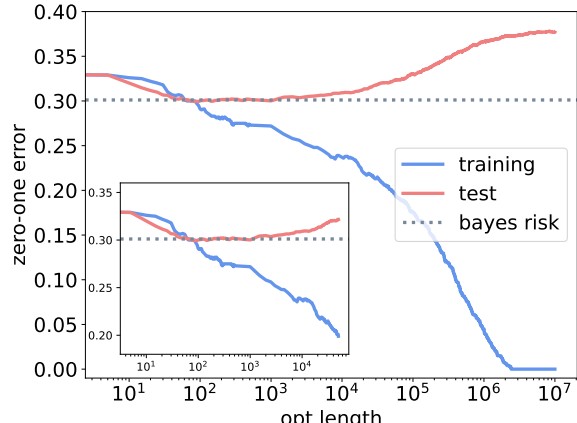

(b) Empirical and population zero-one error

*Figure 1.* The logistic risk and zero-one error along the GD path for an overparameterized logistic regression problem. Here $d = 2000$, $n = 1000$, $\lambda_i = i^{-2}$, $\mathbf{w}^*_{0:100} = 1$ and $\mathbf{w}^*_{100:\infty} = 0$. The optimization length is measured by $\eta t$. The plots show that the excess logistic risk and excess zero-one error are both small for GD with appropriate early stopping, and both grow larger when GD enters the interpolation regime. These demonstrate the regularization of early stopping in GD.

where $\eta > 0$ is a fixed stepsize. We consider zero initialization to simplify the presentation, which does not cause the loss of generality. We aim to compare asymptotic GD, that is, $\mathbf{w}_\infty$, with early-stopped GD, that is, $\mathbf{w}_t$ at a certain finite stopping time $t < \infty$.

**Data model.** We mainly focus on a *well-specified* setting formalized by the following conditions. However, part of our results can also be applied to misspecified cases.

**Assumption 1** (Well-specification). *Let $\mathbf{\Sigma} \in \mathbb{H}^{\otimes 2}$ be positive semi-definite (PSD) and $\mathrm{tr}(\mathbf{\Sigma}) < \infty$. Let $\mathbf{w}^* \in \mathbb{H}$ be such that $\|\mathbf{w}^*\|_{\mathbf{\Sigma}} < \infty$. Assume that $(\mathbf{x}, y) \in \mathbb{H} \otimes \{\pm 1\}$ is given by*

$$\mathbf{x} \sim \mathcal{N}(0, \mathbf{\Sigma}), \quad \Pr(y|\mathbf{x}) = \frac{1}{1 + \exp(-y\mathbf{x}^\top \mathbf{w}^*)}.$$

Under this data model, we have the following standard properties for the logistic risk, zero-one error, and calibration error (see for example (Ji et al., 2021) or Section 4.7 in (Mohri et al., 2018)). The proof is included in Appendix A.1 for completeness.

**Proposition 2.1** (Basic properties). *Under Assumption 1, we have*

A. $\mathbf{w}^* = \arg\min \mathcal{L}(\cdot)$ *and* $\mathbf{w}^* \in \arg\min \mathcal{E}(\cdot)$;

B. *for every* $\mathbf{w} \in \mathbb{H}$*, it holds that*

$$\mathcal{E}(\mathbf{w}) - \min \mathcal{E} \leq 2\sqrt{\mathcal{C}(\mathbf{w})} \leq \sqrt{2} \cdot \sqrt{\mathcal{L}(\mathbf{w}) - \min \mathcal{L}};$$

C. *if additionally we have* $\|\mathbf{w}^*\|_{\mathbf{\Sigma}} \lesssim 1$*, then*

$$\min \mathcal{L} \gtrsim 1, \quad \min \mathcal{E} \gtrsim 1.$$

Proposition 2.1 suggests that the Bayes logistic risk and Bayes zero-one error are attained by the true model parameter $\mathbf{w}^*$. Moreover, the excess zero-one error is controlled by the calibration error, which is further controlled by the excess logistic risk. Thus under Assumption 1, a calibrated estimator is also consistent for classification, and an estimator is calibrated if it attains the Bayes logistic risk asymptotically. However, the reverse might not be true. As we will show later, for overparameterized logistic regression, early-stopped GD is calibrated and consistent for both logistic risk and zero-one error. In contrast, asymptotic GD is poorly calibrated and attains an unbounded logistic risk, although it could be consistent for zero-one error.

**Noise and overparameterization.** Most of our results should be interpreted in the *noisy* and *overparameterized* regime. Specifically, this means

$$\|\mathbf{w}^*\|_{\mathbf{\Sigma}} \lesssim 1 \quad \text{and} \quad \mathrm{rank}(\mathbf{\Sigma}) \geq n.$$

The first condition ensures the population distribution carries a constant amount of noise, as the Bayes logistic risk and Bayes zero-one error are lower bounded by a constant according to Proposition 2.1. In other words, the population distribution is strictly *not* linearly separable. Despite so, the second condition ensures the *linear separability* of the training data almost surely, as the number of effective parameters exceeds the number of training data. In this regime, estimators can *interpolate* the training data, yet this interpolation inherently carries the risk of *overfitting* and *poor calibration*. Our setting aligns well with the prior setting for studying benign overfitting in linear regression (Bartlett et al., 2020; Tsigler & Bartlett, 2023).

**Asymptotic implicit bias.** When the training data is linearly separable (implied by overparameterization), prior works show that GD diverges to infinite in norm while converging in direction to the maximum $\ell_2$-margin direction (Soudry et al., 2018; Ji & Telgarsky, 2018). This characterizes the asymptotic *implicit bias* of GD. See the following proposition for a precise statement.

**Proposition 2.2** (Asymptotic implicit bias). *Assume that* $\text{rank}(\mathbf{x}_1, \ldots, \mathbf{x}_n) \geq n$. *Then the training data* $(\mathbf{x}_i, y_i)_{i=1}^{n}$ *is linearly separable, that is,*

$$\max_{\|\mathbf{w}\|=1} \min_{i \in [n]} y_i \mathbf{x}_i^\top \mathbf{w} > 0.$$

*Let* $\tilde{\mathbf{w}}$ *be the maximum* $\ell_2$-*margin direction, that is,*

$$\tilde{\mathbf{w}} := \arg \max_{\|\mathbf{w}\|=1} \min_{i \in [n]} y_i \mathbf{x}_i^\top \mathbf{w}.$$

*Then* $\tilde{\mathbf{w}}$ *is unique and the following holds for* (GD) *with any stepsize* $\eta > 0$:

$$\|\mathbf{w}_t\| \to \infty, \quad \frac{\mathbf{w}_t}{\|\mathbf{w}_t\|} \to \tilde{\mathbf{w}},$$

*Proof of Proposition 2.2.* Let $\mathbf{X} := (\mathbf{x}_1, \ldots, \mathbf{x}_n)^\top$ and $\mathbf{y} := (y_1, \ldots, y_n)^\top$. Since $\text{rank}(\mathbf{X}) \geq n$, the ordinary least squares estimator

$$\hat{\mathbf{w}} := \mathbf{X}^\top (\mathbf{X}\mathbf{X}^\top)^{-1} \mathbf{y}$$

is well-defined. This implies the linear separability of the training data as $\mathbf{X}\hat{\mathbf{w}} = \mathbf{y}$. If two distinct directions $\tilde{\mathbf{w}}_1$ and $\tilde{\mathbf{w}}_2$ achieve the maximum $\ell_2$-margin, the direction of their average achieves a larger margin, which is a contradiction. So $\tilde{\mathbf{w}}$ must be unique.

For logistic regression with linearly separable data, the implicit bias of GD is established by Soudry et al. (2018); Ji & Telgarsky (2018) when the stepsizes are small such that $\widehat{\mathcal{L}}(\mathbf{w}_t)$ decreases monotonically. The same results can be extended to GD with any fixed stepsize using techniques from (Wu et al., 2023; 2024). $\square$

**Additional notation.** The following notations are handy for presenting our results. Let $(\lambda_i)_{i \geq 1}$ be the eigenvalues of the data covariance $\mathbf{\Sigma}$, sorted in non-increasing order. Let $\mathbf{u}_i$ be the eigenvector of $\mathbf{\Sigma}$ corresponding to $\lambda_i$. Let $(\pi(i))_{i \geq 1}$ be resorted indexes such that $\lambda_{\pi(i)}(\mathbf{u}_{\pi(i)}^\top \mathbf{w}^*)^2$ is non-increasing as a function of $i$. Define

$$\mathbf{w}_{0:k}^* := \sum_{i \leq k} \mathbf{u}_{\pi(i)} \mathbf{u}_{\pi(i)}^\top \mathbf{w}^*, \quad \mathbf{w}_{k:\infty}^* := \sum_{i > k} \mathbf{u}_{\pi(i)} \mathbf{u}_{\pi(i)}^\top \mathbf{w}^*.$$

It is clear that $\|\mathbf{w}^*\|_{\mathbf{\Sigma}} < \infty$ implies that $\|\mathbf{w}_{k:\infty}^*\|_{\mathbf{\Sigma}} = o(1)$ as $k$ increases.

## 3. Upper Bounds for Early-Stopped GD

In this section, we present two risk bounds for early-stopped GD for overparameterized logistic regression and a characterization of the implicit bias of early stopping in GD.

### 3.1. A Bias-Dominating Bound

We first provide a bias-dominating excess logistic risk bound for early-stopped GD in overparameterized logistic regression. The proof is deferred to Appendix B.1.

**Theorem 3.1** (A "bias-dominating" risk bound). *Suppose that Assumption 1 holds. Let $k$ be an arbitrary index. Suppose that the stepsize for* (GD) *satisfies*

$$\eta \leq \frac{1}{C_0(1 + \text{tr}(\mathbf{\Sigma}) + \lambda_1 \ln(1/\delta)/n)},$$

*where* $C_0 > 1$ *is a universal constant. Then with probability at least* $1 - \delta$, *there exists a stopping time* $t$ *such that*

$$\widehat{\mathcal{L}}(\mathbf{w}_t) \leq \widehat{\mathcal{L}}(\mathbf{w}_{0:k}^*) \leq \widehat{\mathcal{L}}(\mathbf{w}_{t-1}).$$

*Moreover, for* (GD) *with this stopping time we have*

$$\mathcal{L}(\mathbf{w}_t) - \min \mathcal{L} \lesssim$$
$$\sqrt{\frac{\max\{1, \text{tr}(\mathbf{\Sigma})\|\mathbf{w}_{0:k}^*\|^2\} \ln^2(n/\delta)}{n}} + \|\mathbf{w}_{k:\infty}^*\|_{\mathbf{\Sigma}}^2.$$

The existence of the desired stopping time is because GD minimizes the empirical risk monotonically (Ji & Telgarsky, 2018). In Theorem 3.1, we choose $k$ to minimize the upper bounds. Intuitively, $k$ determines the number of dimensions in which early-stopped GD is able to learn the true parameter. Moreover, early-stopped GD ignores the remaining dimensions and pays an "approximation" error. A few more remarks on Theorem 3.1 are in order.

**Calibration and consistency.** Theorem 3.1 implies that early-stopped GD attains the Bayes logistic risk asymptotically for *any* logistic regression problem satisfying Assumption 1. To see this, we pick $k$ as an increasing function of $n$ such that $\|\mathbf{w}_{0:k}^*\| = o(n)$. Then $\|\mathbf{w}_{k:\infty}^*\|_{\mathbf{\Sigma}} = o(1)$ since $k$ increases as $n$ increases (recall that $\|\mathbf{w}^*\|_{\mathbf{\Sigma}}$ is finite by Assumption 1). Hence the risk bound in Theorem 3.1 implies that

$$\mathcal{L}(\mathbf{w}_t) - \min \mathcal{L} = o(1) \text{ as } n \text{ increases}.$$

By Proposition 2.1, this also ensures that early-stopped GD induces a conditional probability that approaches the true one and achieves a vanishing excess zero-one error. Hence early-stopped GD is calibrated and consistent for any well-specified logistic regression problem.

As a concrete example, let us consider the following source and capacity conditions (Caponnetto & De Vito, 2007),

$$\lambda_i \asymp i^{-a}, \quad \lambda_i(\mathbf{u}_i^\top \mathbf{w}_i^*)^2 \asymp i^{-b}, \quad a, b > 1. \quad (1)$$

Then Theorem 3.1 implies

$$\mathcal{L}(\mathbf{w}_t) - \min \mathcal{L} = \begin{cases} \tilde{\mathcal{O}}(n^{-1/2}) & b > a + 1, \\ \tilde{\mathcal{O}}(n^{\frac{1-b}{a+b-1}}) & b \leq a + 1. \end{cases}$$

This provides an explicit rate on the excess risk. Note that the obtained rate might not be the sharpest. An improved rate under stronger conditions is provided later in Theorem 3.2. Note that our main purpose here is to show the calibration and consistency of early-stopped GD for every well-specified logistic regression problem.

**Stopping time.** Note that the stopping time $t$ relies on the oracle information of the true parameter $\mathbf{w}^*$. Therefore the "early-stopped GD" in Theorem 3.1 is not a practical algorithm. Instead, we should view Theorem 3.1 as a guarantee for GD with an *optimally tuned* stopping time. It will also be clear later in Section 4 that the optimal stopping time $t$ must be finite for overparameterized logistic regression. Moreover, we point out that the stopping time $t$ is a function of $k$ and thus also depends on the sample size $n$.

Although the stopping time in Theorem 3.1 is implicit, one can compute an upper bound on it using standard optimization and concentration tools. Specifically, GD converges in $\mathcal{O}(1/t)$ rate as the empirical risk is convex and smooth. Moreover, we can compute $\widehat{\mathcal{L}}(\mathbf{w}_{0:k})$ using concentration bounds. These lead to an upper bound on the stopping time.

**Misspecification.** For the simplicity of discussion, we state Theorem 3.1 in a well-specified case formalized by Assumption 1. Nonetheless, from its proof in Appendix B.1, it is clear that the same results also hold in misspecified cases, where we define $\mathbf{w}^* \in \arg\min \mathcal{L}$ and assume $\boldsymbol{\Sigma}^{-1/2}\mathbf{x}$ is subGaussian. Here, we do not need to make assumptions on the true conditional probability $\Pr(y|\mathbf{x})$. In those misspecified cases, however, Proposition 2.1 may not hold. Thus Theorem 3.1 only provides a logistic risk bound but does not yield any bounds on calibration error or zero-one error.

We also note that the proof of Theorem 3.1 can be adapted to other loss functions that are convex, smooth, and Lipschitz.

### 3.2. A Variance-Dominating Bound

From Theorem 3.1, we see that early-stopped GD is consistent and calibrated under the arguably weakest condition on the true parameter, $\|\mathbf{w}^*\|_{\boldsymbol{\Sigma}} < \infty$. However, the attained bound decays at a rate no faster than $\mathcal{O}(1/\sqrt{n})$ as long as $\|\mathbf{w}^*\|_{\boldsymbol{\Sigma}} \gtrsim 1$. In the simpler case where $\|\mathbf{w}^*\| < \infty$, we can tune the stopping time to achieve an improved bound. This is presented in the following theorem. The proof is deferred to Appendix B.2.

**Theorem 3.2** (A "variance-dominating" risk bound)**.** *Suppose that Assumption 1 holds with $\|\mathbf{w}^*\| < \infty$. Let $k$ be an arbitrary index. Suppose that the stepsize for GD satisfies the same condition as in Theorem 3.1 and the stopping time $t$ is such that*

$$\widehat{\mathcal{L}}(\mathbf{w}_t) \leq \widehat{\mathcal{L}}(\mathbf{w}^*) \leq \widehat{\mathcal{L}}(\mathbf{w}_{t-1}).$$

*Assume for simplicity that $\|\mathbf{w}^*\| \gtrsim 1$, $\lambda_1 \lesssim 1$, and $\mathrm{tr}(\boldsymbol{\Sigma}) \gtrsim 1$. Then with probability at least $1 - \delta$, we have*

$$\mathcal{L}(\mathbf{w}_t) - \min \mathcal{L} \lesssim \|\mathbf{w}^*\| \left( \frac{k}{n} + \sqrt{\frac{\sum_{i>k} \lambda_i}{n}} + \frac{\mathrm{tr}(\boldsymbol{\Sigma})^{1/2} \ln\left(n\|\mathbf{w}^*\| \, \mathrm{tr}(\boldsymbol{\Sigma})/\delta\right)}{n} \right).$$

**Comparing Theorems 3.1 and 3.2.** Compared to Theorem 3.1, Theorem 3.2 achieves a faster rate, but is only applicable when $\|\mathbf{w}^*\| < \infty$. Specifically, in the classical finite-dimensional setting where $\|\mathbf{w}^*\| \approx 1$ and $\boldsymbol{\Sigma} = \mathbf{I}_d$, the excess risk bound in Theorem 3.2 decreases at the rate of $\tilde{\mathcal{O}}(d/n)$ while that in Theorem 3.1 decreases at the rate of $\tilde{\mathcal{O}}(\sqrt{d/n})$. For another example, under the source and capacity conditions of (1), Theorem 3.2 provides an improved excess risk bound of $\tilde{\mathcal{O}}(n^{-a/(1+a)})$ when $b > a + 1$, but is not applicable when $b \leq a + 1$.

The stopping time in Theorem 3.1 is designed to handle more general high-dimensional situations that even allow $\|\mathbf{w}^*\| = \infty$. It tends to stop "earlier" so that the bias error tends to dominate the variance error. In comparison, Theorem 3.2 is limited to simpler cases where $\|\mathbf{w}^*\| < \infty$ and sets a "later" stopping time so that the variance error tends to dominate the bias error. Therefore Theorem 3.2 achieves a faster rate.

**Future directions.** Theorems 3.1 and 3.2 are sufficiently powerful for our purpose of demonstrating the benefits of early stopping. However, we point out that neither Theorems 3.1 nor 3.2 reveal the *true* trade-off between the bias and variance errors induced by early stopping. This is unsatisfactory given that in linear regression, the exact trade-off between bias and variance errors has been settled for one-pass SGD (Zou et al., 2023; Wu et al., 2022a;b) and $\ell_2$-regularization (Tsigler & Bartlett, 2023), and has been partially settled for early-stopped GD (Zou et al., 2022, assuming a Gaussian prior). We leave the improvement of these bounds for future work.

From a technical perspective, the gap in analysis between linear regression and logistic regression is significant. All the prior sharp analyses of GD in linear regression make heavy use of explicit calculations with chains of equalities and closed-form solutions. But these fail to hold for GD in logistic regression since the Hessian is no longer fixed. While one might suspect that a limiting analogy can be made where least squares ideas are applied locally around

an optimum, a priori there is no reason to believe that the GD path, which diverges to infinity, even passes near the population optimum, let alone spends a reasonable amount of time there. Moreover, as our lower bounds in Section 4 attest, the GD path exhibits significant curvature. Due to these issues, we believe tools from linear regression can not be merely ported over, and new approaches are required. While we have provided some tools to this end, as above Theorems 3.1 and 3.2 do not tightly characterize the GD path, and much is left to future work.

### 3.3. Implicit Bias of Early Stopping

In this part, we briefly discuss the proof ideas by introducing the following key lemma in our analysis. Variants of this lemma have appeared in (Ji & Telgarsky, 2018; 2019; Shamir, 2021; Telgarsky, 2022; Wu et al., 2024) for analyzing different aspects of GD. For completeness, we include a proof of it in Appendix A.2.

**Lemma 3.3** (Implicit bias of early stopping). *Let $\widehat{\mathcal{L}}(\cdot)$ be convex and $\beta$-smooth. Let $(\mathbf{w}_t)_{t \geq 0}$ be given by* (GD) *with stepsize $\eta \leq 1/\beta$. Then for every $\mathbf{u}$, we have*

$$\frac{\|\mathbf{w}_t - \mathbf{u}\|^2}{2\eta t} + \widehat{\mathcal{L}}(\mathbf{w}_t) \leq \widehat{\mathcal{L}}(\mathbf{u}) + \frac{\|\mathbf{u}\|^2}{2\eta t}, \quad t > 0.$$

This lemma reveals an implicit bias of early-stopping, in which early-stopped GD *attains a small empirical risk while maintaining a relatively small norm*. Specifically, consider a comparator $\mathbf{u}$ and a stopping time $t$ such that

$$\widehat{\mathcal{L}}(\mathbf{w}_t) \leq \widehat{\mathcal{L}}(\mathbf{u}) \leq \widehat{\mathcal{L}}(\mathbf{w}_{t-1}).$$

This stopping time together with Lemma 3.3 (applied to $t - 1$) leads to

$$\widehat{\mathcal{L}}(\mathbf{w}_t) \leq \widehat{\mathcal{L}}(\mathbf{u}), \text{ and } \|\mathbf{w}_{t-1} - \mathbf{u}\| \leq \|\mathbf{u}\|.$$

By optimizing the choice of the comparator $\mathbf{u}$, we see that early-stopped GD achieves a small empirical risk with a relatively small norm.

Besides Lemma 3.3, the remaining efforts for proving Theorems 3.1 and 3.2 are using classical tools of Rademacher complexity (Bartlett & Mendelson, 2002; Kakade et al., 2008) and local Rademacher complexity (Bartlett et al., 2005), respectively. More details can be found in Appendices B.1 and B.2.

Later in Section 5, we will use Lemma 3.3 to show connections between early stopping and $\ell_2$-regularization. We also note that the proof of Theorems 3.1 and 3.2 can be easily adapted to $\ell_2$-regularized empirical risk minimizes.

## 4. Lower Bounds for Interpolating Estimators

In this section, we provide negative results for interpolating estimators by establishing risk lower bounds for them.

### 4.1. Logistic Risk and Calibration Error

The following theorem shows that GD without early stopping must induce an unbounded logistic risk and a positive calibration error in the overparameterized regime. The proof is deferred to Appendix C.1.

**Theorem 4.1** (Lower bounds for logistic risk and calibration error). *Suppose that Assumption 1 holds. Let $\tilde{\mathbf{w}}$ be a unit vector such that $\|\tilde{\mathbf{w}}\|_{\boldsymbol{\Sigma}} > 0$ and let $(\mathbf{w}_t)_{t \geq 0}$ be a sequence of vectors such that*

$$\|\mathbf{w}_t\| \to \infty, \quad \frac{\mathbf{w}_t}{\|\mathbf{w}_t\|} \to \tilde{\mathbf{w}}.$$

*Then we have*

$$\lim_{t \to \infty} \mathcal{C}(\mathbf{w}_t) \geq \exp(-C\|\mathbf{w}^*\|_{\boldsymbol{\Sigma}}), \quad \lim_{t \to \infty} \mathcal{L}(\mathbf{w}_t) = \infty,$$

*where $C > 1$ is a constant.*

Theorem 4.1 shows that for every sequence of estimators that diverges in norm but converges in direction, their induced logistic risk must grow unboundedly and their induced calibration error must be bounded away from zero by a constant. Therefore, their limit is *inconsistent* (for logistic risk) and *poorly calibrated*. According to Proposition 2.2, this applies to GD iterates in the overparameterized regime.

Combining this with our preceding discussion, we see that for *every* well-specified but overparameterized logistic regression problem, GD is calibrated and consistent (for logistic risk) when early stopped, but is poorly calibrated and inconsistent (for logistic risk) at convergence. This contrast demonstrates the benefit of early stopping.

### 4.2. Zero-One Error

The preceding lower bounds in Theorem 4.1 are tied to the divergence of the norm of the estimators. In this part, we show that even when properly normalized, interpolating estimators are still inferior to early-stopped GD. To this end, we consider the zero-one error that is insensitive to the estimator norm. We provide a lower bound on that for interpolating estimators in the next theorem. The proof is deferred to Appendix C.2.

**Theorem 4.2** (A lower bound for zero-one error). *Suppose that Assumption 1 holds. Let $C_2 > C_1 > 1$ be two sufficiently large constants. Assume that $\boldsymbol{\Sigma}^{1/2}\mathbf{w}^*$ is $k$-sparse and $1/C_1 \leq \|\mathbf{w}^*\|_{\boldsymbol{\Sigma}} \leq C_1$. Assume that*

$$n \geq C_1 k \ln(k/\delta), \quad C_1 \leq \frac{\operatorname{rank}(\boldsymbol{\Sigma})}{n \ln(n) \ln(1/\delta)} \leq C_2.$$

*Then with probability at least $1 - \delta$, for every interpolating estimator $\hat{\mathbf{w}}$ such that $\min_{i \in [n]} y_i \mathbf{x}_i^\top \hat{\mathbf{w}} > 0$, we have*

$$\mathcal{E}(\hat{\mathbf{w}}) - \min \mathcal{E} \gtrsim \frac{1}{\sqrt{\ln(n) \ln(1/\delta)}}.$$

Theorem 4.2 characterizes a class of overparameterized logistic regression problems where every interpolating estimator needs at least an *exponential* number of training data to achieve a small excess zero-one error. This applies to asymptotic GD as it converges to the maximum $\ell_2$-margin solution by Proposition 2.2. In contrast, Theorems 3.1 and 3.2 suggests that early-stopped GD can achieve a small excess zero-one error using at most a *polynomial* number of training data under weak conditions. These weak conditions can be, for example, $\|\mathbf{w}^*\| < \infty$ or the sparsity parameter $k$ does not grow with $n$ (see also the examples given by (1)). This separation underscores the benefits of early stopping for reducing sample complexity.

The intuition behind Theorem 4.2 is that there are $k$ informative dimensions and a lot more uninformative dimensions. Since $n \gg k$, the training set cannot be separated purely using the $k$ informative dimensions. Thus, interpolators must use the uninformative dimensions to separate the data, leading to the risk lower bound.

**Future direction.** Note that Theorem 4.2 applies to *every* interpolating estimator. When restricted to the maximum $\ell_2$-margin estimator, the one that GD converges to in direction, we conjecture that a *constant* lower bound on the excess zero-one error can be proved, especially when the spectrum of the data covariance matrix decays fast. This is left for future investigation.

# 5. Early Stopping and $\ell_2$-Regularization

Sections 3 and 4 demonstrate that early stopping carries a certain regularization effect that benefits its statistical performance. This regularization is, however, implicit. In this section, we attempt to provide some intuitions of the implicit regularization of early stopping by establishing its connections to an explicit, $\ell_2$-regularization. An $\ell_2$-regularized *empirical risk minimizer* (ERM) is defined as

$$\mathbf{u}_\lambda := \arg\min_{\mathbf{u}} \widehat{\mathcal{L}}(\mathbf{u}) + \frac{\lambda}{2}\|\mathbf{u}\|^2, \qquad (2)$$

where $\lambda > 0$ is the regularization strength. Note that $\mathbf{u}_\lambda$ is unique and well-defined as long as $\widehat{\mathcal{L}}(\cdot)$ is convex, whereas $\widehat{\mathcal{L}}(\cdot)$ does not have to have a finite minimizer. We refer to $(\mathbf{u}_\lambda)_{\lambda>0}$ given by (2) as the $\ell_2$-*regularization path*. Similarly, we refer to $(\mathbf{w}_t)_{t>0}$ given by (GD) as the *GD path*.

In linear regression, prior works showed that the excess risk of early-stopped GD (and one-pass SGD) is comparable to that of $\ell_2$-regularized ERM (Ali et al., 2019; Zou et al., 2021). For strongly convex and smooth problems, Suggala et al. (2018) provided bounds on the $\ell_2$-distance between the GD and $\ell_2$-regularization paths. In what follows, we establish more connections between the GD and $\ell_2$-regularization paths. We first establish a relative but global

connection in convex (not necessarily strongly convex) and smooth problems, then we establish an asymptotic but absolute connection in overparameterized logistic regression problems.

## 5.1. A Global Connection

The following theorem presents a global comparison of the norm and angle between the GD and $\ell_2$-regularization paths. The proof exploits the implicit regularization results in Lemma 3.3 and is included in Appendix D.1.

**Theorem 5.1** (A global bound). *Let $\widehat{\mathcal{L}}(\cdot)$ be convex and $\beta$-smooth. Consider $(\mathbf{w}_t)_{t\geq0}$ given by (GD) with stepsize $\eta \leq 1/\beta$ and $(\mathbf{u}_\lambda)_{\lambda>0}$ given by (2). Set $\lambda := 1/(\eta t)$. Then we have*

$$\text{for every } t > 0, \quad \|\mathbf{w}_t - \mathbf{u}_\lambda\| \leq \frac{1}{\sqrt{2}}\|\mathbf{w}_t\|.$$

*As a direct consequence, the following holds for every $t > 0$:*

$$\cos(\mathbf{w}_t, \mathbf{u}_\lambda) \geq \frac{1}{\sqrt{2}},$$

$$\frac{\sqrt{2}}{1+\sqrt{2}}\|\mathbf{u}_\lambda\| \leq \|\mathbf{w}_t\| \leq \frac{\sqrt{2}}{\sqrt{2}-1}\|\mathbf{u}_\lambda\|.$$

Theorem 5.1 establish a global but relative connection between the GD and $\ell_2$-regularization paths for all convex and smooth problems. Specifically, starting from the same zero initialization, the angle between the two paths is no more than $\pi/4$, and the norm of the two paths differs by a factor within $0.585$ and $3.415$. We point out this relative connection holds *globally* for every stopping time (with its corresponding regularization strength) and for every convex and smooth problem. In particular, it applies to overparameterized logistic regression, which is smooth and convex but not strongly convex. We also note that using the norm bounds in Theorem 5.1, the upper bounds in Theorems 3.1 and 3.2 for early-stopped GD can be easily adapted to $\ell_2$-regularized ERM.

Theorem 5.1 cannot be improved without making further assumptions. This is because the GD and $\ell_2$-regularization paths could converge to two distinct limits (as $t \to \infty$ and $\lambda \to 0$) in convex but non-strongly convex problems (see Suggala et al., 2018, Section 4). So in general, we cannot expect their distance to be small in the absolute sense.

## 5.2. An Asymptotic Comparison

We have established a global but relative connection between the GD and $\ell_2$-regularization paths in Theorem 5.1. We now turn to logistic regression with linearly separable data and establish an absolute but asymptotic connection between the two paths.

In logistic regression with linearly separable data, both GD and $\ell_2$-regularization paths diverge to infinite in norm (as $t \to \infty$ and $\lambda \to 0$) while converging in direction to the maximum $\ell_2$-margin solution (Rosset et al., 2004; Soudry et al., 2018; Ji & Telgarsky, 2018; Ji et al., 2020). Therefore their angle tends to zero asymptotically (Suggala et al., 2018; Ji et al., 2020). This characterization is more precise than the $\pi/4$ global angle bound from Theorem 5.1.

However, it remains unclear how the $\ell_2$-distance between the two paths evolves in logistic regression with linearly separable data. Quite surprisingly, we will show that their $\ell_2$-distance tends to zero under a widely used condition (Soudry et al., 2018; Ji & Telgarsky, 2021; Wu et al., 2023), but could diverge to infinity in the worst case.

Let $\mathbf{X} := (\mathbf{x}_1, \ldots, \mathbf{x}_n)^\top$ and $\mathbf{y} := (y_1, \ldots, y_n)^\top$ be a set of linearly separable data. Then the Lagrangian dual of the margin maximization program in Proposition 2.2 is given by (see Hsu et al., 2021, for example)

$$\max_{\boldsymbol{\beta} \in \mathbb{R}^n} \; -\frac{1}{2}\boldsymbol{\beta}^\top \mathbf{X}\mathbf{X}^\top \boldsymbol{\beta} + \boldsymbol{\beta}^\top \mathbf{y} \quad \text{s.t.} \; y_i\boldsymbol{\beta}_i \geq 0, \; i \in [n].$$

Here, $\boldsymbol{\beta}$ are the dual variables multiplied by $\mathbf{y}$ entry-wise. Let $\hat{\boldsymbol{\beta}}$ be the solution to the above problem. Let $\mathcal{S}_+ := \{i \in [n] : y_i\hat{\boldsymbol{\beta}}_i > 0\}$ be the set of support vectors (with strictly positive dual variables). The following condition assumes the coverage of the support vectors.

**Assumption 2** (Support vectors condition). *Assume that* $\operatorname{rank}\{\mathbf{x}_i : i \in \mathcal{S}_+\} = \operatorname{rank}\{\mathbf{x}_i : i \in [n]\}$.

Assumption 2 has been widely used in the analysis of the implicit bias (Soudry et al., 2018; Ji & Telgarsky, 2021; Wu et al., 2023). In particular, Assumption 2 holds if every data is a support vector, which is common in high-dimensional situations (Hsu et al., 2021; Wang & Thrampoulidis, 2022; Cao et al., 2021).

The following theorem provides an asymptotic bound on the $\ell_2$-distance between the GD and $\ell_2$-regularization paths under Assumption 2. The proof is deferred to Appendix D.2.

**Theorem 5.2** (An asymptotic bound). *Let* $(\mathbf{x}_i, y_i)_{i=1}^n$ *be a linearly separable dataset that satisfies Assumption 2. Let* $(\mathbf{w}_t)_{t>0}$ *and* $(\mathbf{u}_\lambda)_{\lambda>0}$ *be the GD and* $\ell_2$*-regularization paths, respectively, for logistic regression with* $(\mathbf{x}_i, y_i)_{i=1}^n$. *Then there exists* $\lambda$ *as a function of* $t$ *such that*

$$\|\mathbf{w}_t - \mathbf{u}_{\lambda(t)}\| \to 0, \quad \text{while} \quad \|\mathbf{w}_t\|, \|\mathbf{u}_{\lambda(t)}\| \to \infty,$$

*as* $t \to \infty$.

For logistic regression with linearly separable data under Assumption 2, Theorem 5.2 shows that the $\ell_2$-distance between the GD and $\ell_2$-regularization paths tends to zero, despite that both paths diverge to infinity in their norm. Note

that this implies their angle converges to zero, and is more precise than the relative norm bound from Theorem 5.1.

However, this sharp asymptotic connection is strongly tied to Assumption 2. Surprisingly, when Assumption 2 fails to hold, the $\ell_2$-distance between the GD and $\ell_2$-regularization paths could tend to infinity instead. This is shown in the following theorem. The proof is deferred to Appendix D.3.

**Theorem 5.3** (A counter example). *Consider the following dataset*

$$\mathbf{x}_1 := (\gamma, 0)^\top, \; y_1 := 1, \; \mathbf{x}_2 := (\gamma, \gamma_2)^\top, \; y_2 := 1,$$

*where* $0 < \gamma_2 < \gamma < 1$. *Then* $(\mathbf{x}_i, y_i)_{i=1,2}$ *is linearly separable but violates Assumption 2. Let* $(\mathbf{w}_t)_{t \geq 0}$ *and* $(\mathbf{u}_\lambda)_{\lambda \geq 0}$ *be the GD and* $\ell_2$*-regularization paths respectively for logistic regression with* $(\mathbf{x}_i, y_i)_{i=1,2}$. *Then* $\|\mathbf{w}_t\| \to \infty$ *as* $t \to \infty$. *Moreover, for* every *map* $\lambda : \mathbb{R}_{\geq 0} \to \mathbb{R}_{\geq 0}$, *we have*

$$\|\mathbf{w}_t - \mathbf{u}_{\lambda(t)}\| \gtrsim \ln\ln(\|\mathbf{w}_t\|) \to \infty.$$

This simple yet strong counter-example suggests that the $\ell_2$-distance between the GD and $\ell_2$-regularization path can diverge to infinity when Assumption 2 fails to hold.

**Future directions.** We conjecture that for logistic regression with linearly separable data, the limit of the $\ell_2$-distance between the GD and $\ell_2$-regularized paths is *either zero or infinity*, and the phase transition is determined by a certain geometric property of the dataset (for example, Assumption 2). The reasoning behind this conjecture is as follows. Note that Assumption 2 implies that the dataset projected perpendicular to the max-margin direction (called "projected dataset") is strictly nonseparable (Wu et al., 2023, Lemma 3.1). This is the main property used in Theorem 5.2. Moreover, in Theorem 5.3, the "projected dataset" is nonseparable but with margin zero—we conjecture this property is sufficient for Theorem 5.3 to hold. Now for a generic separable dataset, we check the "projected dataset": if it is strictly nonseparable, Theorem 5.2 holds; if it is nonseparable but with margin zero, we conjecture Theorem 5.3 holds; otherwise, it is separable (with positive margin), we decompose the dataset recursively. This is the reasoning behind our conjecture.

It also remains unclear to what extent early stopping replicates the effects of explicit regularization for logistic regression. Specifically, is there a logistic regression example such that early-stopped GD has a better calibration/logistic risk rate than $\ell_2$-regularization or vice-versa? This is left for future investigation, as our current bounds are not sharp enough to yield a concrete answer.

## 6. Related Works

We now discuss additional related works.

**Benign overfitting in logistic regression.** A line of work shows the benign overfitting of the asymptotic GD (or the maximum $\ell_2$-margin estimator) in overparameterized logistic regression under a variety of assumptions (Montanari et al., 2019; Chatterji & Long, 2021; Cao et al., 2021; Wang & Thrampoulidis, 2022; Muthukumar et al., 2021; Shamir, 2023). Our results are not a violation of theirs, instead, we show an additional regularization of early-stopping, which brings statistical advantages of early-stopped GD over asymptotic GD such as calibration and a smaller sample complexity.

**M-estimators for logistic regression.** In the classical finite $d$-dimensional setting, the sample complexity of the *empirical risk minimizer* (ERM) for logistic regression is well-studied (Ostrovskii & Bach, 2021; Kuchelmeister & van de Geer, 2024; Hsu & Mazumdar, 2024; Chardon et al., 2024), where the minimax rate is known to be $\mathcal{O}(d/n)$. Different from theirs, we focus on an overparameterized regime, where the ERM of logistic regression does not even exist. When specialized to their setting, our Theorem 3.2 recovers the comparable $\tilde{\mathcal{O}}(d/n)$ rate.

In the nonparametric setting, the works by (Bach, 2010; Marteau-Ferey et al., 2019) provided logistic risk bounds for $\ell_2$-regularized ERM. Bach (2010) only considered a fixed design setting, whereas Marteau-Ferey et al. (2019) required that $\|\mathbf{w}^*\| < \infty$. Different from theirs, we aim to understand the benefits of the implicit regularization of early-stopping, instead of that of explicit $\ell_2$-regularization. Moreover, we show that early-stopped GD achieves a vanishing excess logistic risk as long as $\|\mathbf{w}^*\|_{\boldsymbol{\Sigma}} < \infty$, without assuming a finite $\|\mathbf{w}^*\|$. In the regimes where our results are directly comparable, however, our risk bounds might be less tight than theirs. We leave it as a future work to improve our current bounds.

The work by Bach (2014) considered one-pass SGD for logistic regression assuming strong convexity around the true model parameter. This strong convexity assumption, however, is prohibitive in our high-dimensional settings.

There is a line of works (Sur & Candès, 2019; Candès & Sur, 2020) focused on the existence of ERM for logistic regression in a propotional limit setting (assuming that $n, d \to \infty$ in a fixed ratio, see also (Chardon et al., 2024) in the finite-dimensional setting). This is quite apart from our focus, where ERM never exists due to overparameterization.

**Separable distribution.** There are logistic risk bounds of early-stopped GD (and one-pass SGD) developed in the *noiseless* cases, assuming a separable population distribution (Ji & Telgarsky, 2018; Shamir, 2021; Telgarsky, 2022; Schliserman & Koren, 2024). These results do not imply any benefits of early stopping, as their setting is noiseless. In comparison, we consider overparameterized logistic regression with a strictly non-separable population distribution, where the risk of overfitting is prominent. In this case, our results suggest that early stopping plays a significant role in preventing overfitting.

**Early stopping for classification.** In the boosting literature, an early work by Zhang & Yu (2005) showed that boosting methods (that can be interpreted as coordinate descent) with early stopping are consistent in the classification sense; related refined studies for boosting with the squared loss with early stopping were also provided by Bühlmann & Yu (2003). The paper is also notable for giving the first proof of boosting methods converging to the maximum margin solution (Zhang & Yu, 2005, Appendix D), which was later refined with rates by (Telgarsky, 2013). Their results can be converted to GD. In particular, related concepts were used to prove consistency of early-stopped GD for shallow networks in the lazy regime (Ji et al., 2021). In contrast with the present work that focuses on high-dimensional cases, the preceding works only deal with finite-dimensional settings. Moreover, none of those works provide lower bounds for interpolating estimators and tight links to the regularization path which are provided in the present work.

**Classification calibration.** Proposition 2.1 captures a very nice consequence of logistic loss minimization: *calibration* and *classification-calibration*, respectively recovery of the optimal conditional probability model and of the optimal classifier. For more general convex losses, the ability to construct a general conditional probability model was developed by Zhang (2004) as a conceptual tool in establishing classification calibration, but without explicitly controlling calibration error. A further abstract treatment of classification calibratoin was later presented by Bartlett et al. (2006). The refined statistical rates, separations, and early-stopping consequences studied in the present work were not considered in those works.

## 7. Conclusion

We show the benefits of early stopping in GD for overparameterized and well-specified logistic regression. We show that for every well-specified logistic regression problem, early-stopped GD is calibrated while asymptotic GD is not. Furthermore, we show that early-stopped GD achieves a small excess zero-one error with only a polynomial number of samples, in contrast to interpolating estimators, including asymptotic GD, which require an exponential number of samples to achieve the same. Finally, we establish nonasymptotic bounds on the differences between the GD and the $\ell_2$-regularization paths. Altogether, we underscore the statistical benefits of early stopping, partially explained by its connection with $\ell_2$-regularization.

## Acknowledgments

We gratefully acknowledge the support of the NSF for FODSI through grant DMS-2023505, of the NSF and the Simons Foundation for the Collaboration on the Theoretical Foundations of Deep Learning through awards DMS-2031883 and #814639, of the NSF through grants DMS-2209975 and DMS-2413265, and of the ONR through MURI award N000142112431. The authors are also grateful to the Simons Institute for hosting them during parts of this work.

## Impact Statement

This paper presents work whose goal is to advance the field of Machine Learning. There are many potential societal consequences of our work, none of which we feel must be specifically highlighted here.

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

# A. Basic Results

## A.1. Proof of Proposition 2.1

*Proof of Proposition 2.1.* We first compute the logistic loss. Define

$$p_{\mathbf{x}}(\mathbf{w}) := \frac{1}{1 + \exp(-\mathbf{x}^\top \mathbf{w})}, \quad p_{\mathbf{x}}^* := \frac{1}{1 + \exp(-\mathbf{x}^\top \mathbf{w}^*)}.$$

Then under Assumption 1 we have

$$\begin{aligned}
\mathcal{L}(\mathbf{w}) &= \mathbb{E}_{\mathbf{x}, y} \ln(1 + \exp(-y\mathbf{x}^\top \mathbf{w})) \\
&= -\mathbb{E}_{\mathbf{x}} \Big( p_{\mathbf{x}}^* \ln p_{\mathbf{x}}(\mathbf{w}) + (1 - p_{\mathbf{x}}^*) \ln \big(1 - p_{\mathbf{x}}(\mathbf{w})\big) \Big) \\
&= \mathbb{E}_{\mathbf{x}} \big( H(p_{\mathbf{x}}^*) + \mathrm{KL}\left(p_{\mathbf{x}}^* \,\|\, p_{\mathbf{x}}(\mathbf{w})\right) \big),
\end{aligned}$$

where the first term is the entropy of a Bernoulli distribution with a head probability of $p_{\mathbf{x}}^*$, and the second term is the KL divergence between two Bernoulli distributions with head probabilities of $p_{\mathbf{x}}^*$ and $p_{\mathbf{x}}(\mathbf{w})$, respectively. It is then clear that $\mathbf{w}^*$ is the unique minimizer of $\mathcal{L}(\mathbf{w})$.

We then compute the zero-one error under Assumption 1. A similar calculation can be found in, for example, Lemma 4.5 in (Mohri et al., 2018). Note that $\mathcal{E}(0) = 1$. If $\mathbf{w}^* = 0$, then for every $\mathbf{w}$,

$$\mathcal{E}(\mathbf{w}) = \mathbb{E}_{\mathbf{x}, y} \mathbb{1}\left[y\mathbf{x}^\top \mathbf{w} \le 0\right] = \mathbb{E}_{\mathbf{x}} 0.5\big(\mathbb{1}\left[\mathbf{x}^\top \mathbf{w} \le 0\right] + \mathbb{1}\left[\mathbf{x}^\top \mathbf{w} \ge 0\right]\big) \ge 1 = \mathcal{E}(\mathbf{w}^*),$$

so $\mathbf{w}^* \in \arg\min \mathcal{E}(\cdot)$. If $\mathbf{w}^* \ne 0$, we have $\mathbf{x}^\top \mathbf{w}^* = 0$ is measure zero. Then we have

$$\mathcal{E}(\mathbf{w}^*) = \mathbb{E}_{\mathbf{x}} \big( p_{\mathbf{x}}^* \mathbb{1}\left[\mathbf{x}^\top \mathbf{w}^* \le 0\right] + (1 - p_{\mathbf{x}}^*) \mathbb{1}\left[\mathbf{x}^\top \mathbf{w}^* \ge 0\right] \big) = \mathbb{E}_{\mathbf{x}} \min\{p_{\mathbf{x}}^*, 1 - p_{\mathbf{x}}^*\} \le 0.5 < \mathcal{E}(0).$$

It remains to check that $\mathcal{E}(\mathbf{w}^*) \le \mathcal{E}(\mathbf{w})$ for all $\mathbf{w} \ne 0$. When $\mathbf{w}^*$ and $\mathbf{w}$ are both non-zero, we have $\mathbf{x}^\top \mathbf{w}^* = 0$ and $\mathbf{x}^\top \mathbf{w} = 0$ are measure zero, then we have

$$\begin{aligned}
\mathcal{E}(\mathbf{w}) - \mathcal{E}(\mathbf{w}^*) &= \mathbb{E}_{\mathbf{x}, y} \mathbb{1}\left[y\mathbf{x}^\top \mathbf{w} \le 0\right] - \mathbb{1}\left[y\mathbf{x}^\top \mathbf{w}^* \le 0\right] \\
&= \mathbb{E}_{\mathbf{x}} p_{\mathbf{x}}^* \big(\mathbb{1}\left[\mathbf{x}^\top \mathbf{w} \le 0\right] - \mathbb{1}\left[\mathbf{x}^\top \mathbf{w}^* \le 0\right]\big) + (1 - p_{\mathbf{x}}^*)\big(\mathbb{1}\left[\mathbf{x}^\top \mathbf{w} \ge 0\right] - \mathbb{1}\left[\mathbf{x}^\top \mathbf{w}^* \ge 0\right]\big) \\
&= \mathbb{E}_{\mathbf{x}} p_{\mathbf{x}}^* \big(\mathbb{1}\left[\mathbf{x}^\top \mathbf{w} \le 0\right] - \mathbb{1}\left[\mathbf{x}^\top \mathbf{w}^* \le 0\right]\big) + (1 - p_{\mathbf{x}}^*)\big(\mathbb{1}\left[\mathbf{x}^\top \mathbf{w}^* < 0\right] - \mathbb{1}\left[\mathbf{x}^\top \mathbf{w} < 0\right]\big) \\
&= \mathbb{E}_{\mathbf{x}} (2p_{\mathbf{x}}^* - 1)\big(\mathbb{1}\left[\mathbf{x}^\top \mathbf{w} \le 0\right] - \mathbb{1}\left[\mathbf{x}^\top \mathbf{w}^* \le 0\right]\big) \\
&= \mathbb{E}_{\mathbf{x}} (2p_{\mathbf{x}}^* - 1)\mathbb{1}\left[\mathbf{x}^\top \mathbf{w} \le 0\right] \mathbb{1}\left[\mathbf{x}^\top \mathbf{w}^* > 0\right] + (1 - 2p_{\mathbf{x}}^*)\mathbb{1}\left[\mathbf{x}^\top \mathbf{w} > 0\right] \mathbb{1}\left[\mathbf{x}^\top \mathbf{w}^* \le 0\right] \\
&= \mathbb{E}_{\mathbf{x}} |2p_{\mathbf{x}}^* - 1|\mathbb{1}\left[\mathbf{x}^\top \mathbf{w}\mathbf{x}^\top \mathbf{w}^* \le 0\right],
\end{aligned}$$

which is always non-negative. Therefore we have $\mathbf{w}^* \in \arg\min \mathcal{E}(\mathbf{w})$. We complete the proof of the first claim.

The second claim follows from the following. The calculation is similar to the proof of Theorem 4.7 in (Mohri et al., 2018).

$$\begin{aligned}
\mathcal{E}(\mathbf{w}) - \mathcal{E}(\mathbf{w}^*) &= 2\mathbb{E}_{\mathbf{x}} |p_{\mathbf{x}}^* - 1/2|\mathbb{1}\left[\mathbf{x}^\top \mathbf{w}\mathbf{x}^\top \mathbf{w}^* \le 0\right] \\
&\le 2\mathbb{E}_{\mathbf{x}} |p_{\mathbf{x}}^* - p_{\mathbf{x}}(\mathbf{w})|\mathbb{1}\left[\mathbf{x}^\top \mathbf{w}\mathbf{x}^\top \mathbf{w}^* \le 0\right] \\
&\le 2\mathbb{E}_{\mathbf{x}} |p_{\mathbf{x}}^* - p_{\mathbf{x}}(\mathbf{w})| \\
&\le 2\sqrt{\mathbb{E}_{\mathbf{x}} |p_{\mathbf{x}}^* - p_{\mathbf{x}}(\mathbf{w})|^2} \\
&\le 2\sqrt{\frac{1}{2}\mathbb{E}_{\mathbf{x}}\mathrm{KL}\left(p_{\mathbf{x}}^* \,\|\, p_{\mathbf{x}}(\mathbf{w})\right)} \\
&= \sqrt{2} \cdot \sqrt{L(\mathbf{w}) - L(\mathbf{w}^*)},
\end{aligned}$$

where the last inequality is by Pinsker's inequality. We complete the proof of the second claim.

We now prove the third claim under Assumption 1. Notice that

$$\mathcal{L}(\mathbf{w}) = \mathbb{E} \ln(1 + \exp(-y\mathbf{x}^\top \mathbf{w})) \ge \mathbb{E} \ln(2)\mathbb{1}\left[y\mathbf{x}^\top \mathbf{w} \le 0\right] = \ln(2) \cdot \mathcal{E}(\mathbf{w}), \quad \text{for all } \mathbf{w}.$$

Therefore a lower bound on $\mathcal{E}(\mathbf{w}^*)$ implies a lower bound on $\mathcal{L}(\mathbf{w}^*)$. We lower bound $\mathcal{E}(\mathbf{w}^*)$ by

$$
\begin{aligned}
\mathcal{E}(\mathbf{w}^*) &= \mathbb{E}\mathbb{1}\left[y\mathbf{x}^\top\mathbf{w}^* \leq 0\right] \\
&= \mathbb{E}\mathbb{1}\left[\mathbf{x}^\top\mathbf{w}^* \leq 0\right]\frac{1}{1+\exp(-\mathbf{x}^\top\mathbf{w}^*)} + \mathbb{1}\left[\mathbf{x}^\top\mathbf{w}^* \geq 0\right]\frac{1}{1+\exp(\mathbf{x}^\top\mathbf{w}^*)} \\
&= 2\mathbb{E}_{g\sim\mathcal{N}(0,1)}\frac{1}{1+\exp(g\|\mathbf{w}^*\|_{\boldsymbol{\Sigma}})}\mathbb{1}\left[g \geq 0\right] \\
&= \frac{\sqrt{2}}{\pi}\int_0^\infty \frac{\exp(-g^2/2)}{1+\exp(g\|\mathbf{w}^*\|_{\boldsymbol{\Sigma}})}\mathrm{d}g \\
&\geq \frac{\sqrt{2}}{2\pi}\int_0^\infty \exp(-g^2/2)\exp(-g\|\mathbf{w}^*\|_{\boldsymbol{\Sigma}})\mathrm{d}g \\
&= \frac{\sqrt{2}}{2\pi}\cdot\exp\left(\|\mathbf{w}^*\|_{\boldsymbol{\Sigma}}^2/2\right)\int_0^\infty \exp\left(-(g+\|\mathbf{w}^*\|_{\boldsymbol{\Sigma}})^2/2\right)\mathrm{d}g \\
&\geq \frac{\sqrt{2}}{2\pi}\cdot\frac{\sqrt{2}}{\|\mathbf{w}^*\|_{\boldsymbol{\Sigma}}/\sqrt{2}+\sqrt{\|\mathbf{w}^*\|_{\boldsymbol{\Sigma}}^2/2+2}} \\
&\geq \frac{1}{\sqrt{2}\pi(\|\mathbf{w}^*\|_{\boldsymbol{\Sigma}}+1)},
\end{aligned}
$$

where we use the following error bounds (Abramowitz & Stegun, 1965),

$$
\frac{1}{x+\sqrt{x^2+2}} \leq e^{x^2}\int_x^\infty e^{-t^2}\mathrm{d}t \leq \frac{1}{x+\sqrt{x^2+4/\pi}}, \quad x \geq 0.
$$

This completes the proof. □

## A.2. Proof of Lemma 3.3

*Proof of Lemma 3.3.* For $\eta \leq 1/\beta$, we have the descent lemma, that is,

$$
\widehat{\mathcal{L}}(\mathbf{w}_{t+1}) \leq \widehat{\mathcal{L}}(\mathbf{w}_t) - \eta\|\nabla\widehat{\mathcal{L}}(\mathbf{w}_t)\|^2 + \frac{\eta^2\beta}{2}\|\nabla\widehat{\mathcal{L}}(\mathbf{w}_t)\|^2 \leq \widehat{\mathcal{L}}(\mathbf{w}_t) - \frac{\eta}{2}\|\nabla\widehat{\mathcal{L}}(\mathbf{w}_t)\|^2.
$$

Then for the quadratic potential centered at $\mathbf{u}$, we have

$$
\begin{aligned}
\|\mathbf{w}_{t+1}-\mathbf{u}\|^2 &= \|\mathbf{w}_t-\mathbf{u}\|^2 + 2\eta\langle\nabla\widehat{\mathcal{L}}(\mathbf{w}_t),\mathbf{u}-\mathbf{w}_t\rangle + \eta^2\|\nabla\widehat{\mathcal{L}}(\mathbf{w}_t)\|^2 \\
&\leq \|\mathbf{w}_t-\mathbf{u}\|^2 + 2\eta\left(\widehat{\mathcal{L}}(\mathbf{u})-\widehat{\mathcal{L}}(\mathbf{w}_t)\right) + 2\eta\left(\widehat{\mathcal{L}}(\mathbf{w}_t)-\widehat{\mathcal{L}}(\mathbf{w}_{t+1})\right) \\
&= \|\mathbf{w}_t-\mathbf{u}\|^2 + 2\eta\left(\widehat{\mathcal{L}}(\mathbf{u})-\widehat{\mathcal{L}}(\mathbf{w}_{t+1})\right),
\end{aligned}
$$

where the inequality is due to the convexity and the descent lemma. Telescoping the sum and rearranging, we have

$$
\frac{\|\mathbf{w}_t-\mathbf{u}\|^2}{2\eta t} + \frac{1}{t}\sum_{k=1}^t \widehat{\mathcal{L}}(\mathbf{w}_k) \leq \widehat{\mathcal{L}}(\mathbf{u}) + \frac{\|\mathbf{w}_0-\mathbf{u}\|^2}{2\eta t}.
$$

Using the descent lemma and the initial condition $\mathbf{w}_0 = 0$, we have

$$
\frac{\|\mathbf{w}_t-\mathbf{u}\|^2}{2\eta t} + \widehat{\mathcal{L}}(\mathbf{w}_t) \leq \widehat{\mathcal{L}}(\mathbf{u}) + \frac{\|\mathbf{u}\|^2}{2\eta t}.
$$

This completes the proof. □

# B. Upper Bounds for Early-Stopped GD

## B.1. Proof of Theorem 3.1

**Lemma B.1.** *Let $\beta := C_0\left(1+\mathrm{tr}(\boldsymbol{\Sigma})+\lambda_1\ln(1/\delta)/n\right)$, where $C_0 > 1$ is a sufficiently large constant. Assume that $\eta \leq 1/\beta$ and $t$ is such that $\widehat{\mathcal{L}}(\mathbf{w}_{0:k}^*) \leq \widehat{\mathcal{L}}(\mathbf{w}_{t-1})$. Then with probability at least $1-\delta$, we have $\|\mathbf{w}_t - \mathbf{w}_{0:k}^*\| \leq 1 + \|\mathbf{w}_{0:k}\|$.*

*Proof of Lemma B.1.* We first show that the following holds with probability at least $1 - \delta$:

$$\left\|\nabla\widehat{\mathcal{L}}(\mathbf{w})\right\| \le \sqrt{\beta}, \quad \left\|\nabla^2\widehat{\mathcal{L}}(\mathbf{w})\right\| \le \beta.$$

This is because

$$\left\|\nabla\widehat{\mathcal{L}}(\mathbf{w})\right\| = \left\|\frac{1}{n}\sum_{i=1}^{n}\ell'(y_i\mathbf{x}_i^\top\mathbf{w})y_i\mathbf{x}_i\right\| \le \frac{1}{n}\sum_{i=1}^{n}\|\mathbf{x}_i\| \le \sqrt{\frac{1}{n}\sum_{i=1}^{n}\|\mathbf{x}_i\|^2},$$

and

$$\left\|\nabla^2\widehat{\mathcal{L}}(\mathbf{w})\right\| = \left\|\frac{1}{n}\sum_{i=1}^{n}\ell''(y_i\mathbf{x}_i^\top\mathbf{w})\mathbf{x}_i\mathbf{x}_i^\top\right\| \le \frac{1}{n}\sum_{i=1}^{n}\|\mathbf{x}_i\|^2.$$

So it remains to bound $\sum_{i=1}^{n}\|\mathbf{x}_i\|^2$. Let $z_{ij}$'s be independent Gaussian random variables, then by Assumption 1 and Bernstein's inequality, we have the following with probability at least $1 - \delta$:

$$\sum_{i=1}^{n}\|\mathbf{x}_i\|^2 = \sum_{i=1}^{n}\sum_{j}\lambda_j z_{ij}^2 \le n\operatorname{tr}(\Sigma) + C_1\left(\sqrt{n\sum_j\lambda_j^2\ln(1/\delta)} + \lambda_1\ln(1/\delta)\right)$$

$$\le C_0\big(n\operatorname{tr}(\Sigma) + \lambda_1\ln(1/\delta)\big) \le \beta,$$

where $C_0, C_1 > 1$ are constants.

So far we have shown $\widehat{\mathcal{L}}$ is $\beta$-smooth and $\sqrt{\beta}$-Lipschitz for $\beta > 1$. By the stopping criterion, we have $\widehat{\mathcal{L}}(\mathbf{w}_{0:k}^*) \le \widehat{\mathcal{L}}(\mathbf{w}_{t-1})$. Then by applying Lemma 3.3 to $\mathbf{w}_{t-1}$ and $\mathbf{u} = \mathbf{w}_{0:k}^*$, we have

$$\|\mathbf{w}_{t-1} - \mathbf{w}_{0:k}^*\|^2 \le 2\eta(t-1)\big(\widehat{\mathcal{L}}(\mathbf{w}_{0:k}^*) - \widehat{\mathcal{L}}(\mathbf{w}_{t-1})\big) + \|\mathbf{w}_{0:k}^*\|^2 \le \|\mathbf{w}_{0:k}^*\|^2$$

$$\Rightarrow \quad \|\mathbf{w}_{t-1} - \mathbf{w}_{0:k}^*\| \le \|\mathbf{w}_{0:k}^*\|.$$

Then by the Lipschitzness, we get

$$\|\mathbf{w}_t - \mathbf{w}_{0:k}^*\| \le \|\mathbf{w}_t - \mathbf{w}_{t-1}\| + \|\mathbf{w}_{t-1} - \mathbf{w}_{0:k}^*\| \le \eta\sqrt{\beta} + \|\mathbf{w}_{0:k}^*\| \le \frac{1}{\sqrt{\beta}} + \|\mathbf{w}_{0:k}^*\| \le 1 + \|\mathbf{w}_{0:k}^*\|,$$

where we use $\left\|\nabla\widehat{\mathcal{L}}(\mathbf{w})\right\| \le \sqrt{\beta}$, $\eta \le 1/\beta$, and $\beta \ge 1$. This completes the proof. $\qquad\square$

**Lemma B.2.** *Let* $\mathbf{w}^* \in \arg\min \mathcal{L}(\mathbf{w})$, *then for every* $\mathbf{w}$, *we have*

$$\mathcal{L}(\mathbf{w}) \le \mathcal{L}(\mathbf{w}^*) + \frac{1}{2}\|\mathbf{w} - \mathbf{w}^*\|_{\Sigma}^2.$$

*Proof of Lemma B.2.* Notice that

$$\nabla^2\mathcal{L}(\mathbf{w}) = \mathbb{E}\ell''(y\mathbf{x}^\top\mathbf{w})\mathbf{x}\mathbf{x}^\top = \mathbb{E}\frac{\mathbf{x}\mathbf{x}^\top}{\big(1 + \exp(\mathbf{x}^\top\mathbf{w})\big)\big(1 + \exp(-\mathbf{x}^\top\mathbf{w})\big)} \preceq \mathbb{E}\mathbf{x}\mathbf{x}^\top = \Sigma.$$

Moreover, we have $\nabla\mathcal{L}(\mathbf{w}^*) = 0$. Then by the midpoint theorem, there exists a $\mathbf{v}$ such that

$$\mathcal{L}(\mathbf{w}) - \mathcal{L}(\mathbf{w}^*) = \langle\nabla\mathcal{L}(\mathbf{w}^*), \mathbf{w} - \mathbf{w}^*\rangle + \frac{1}{2}(\mathbf{w} - \mathbf{w}^*)^\top\nabla^2\mathcal{L}(\mathbf{v})(\mathbf{w} - \mathbf{w}^*) \le \frac{1}{2}\|\mathbf{w} - \mathbf{w}^*\|_{\Sigma}^2.$$

This completes the proof. $\qquad\square$

**Lemma B.3.** *Let* $C_1 > 1$ *be a sufficiently large constant. Then with probability at least* $1 - \delta$,

$$\sup_{\|\mathbf{w}\|\le W}|\mathcal{L}(\mathbf{w}) - \widehat{\mathcal{L}}(\mathbf{w})| \le C_1 W\sqrt{\frac{(1 + \operatorname{tr}(\Sigma))\ln(n/\delta)\ln(1/\delta)}{n}}.$$

*Proof of Lemma B.3.* This is by standard Rademacher complexity arguments for 1-Lipschitz loss and linear function class (Bartlett & Mendelson, 2002; Kakade et al., 2008).

Similarly to the proof of Theorem 3.2, let $\tilde{\mathbf{x}} := \mathbf{x}\mathbb{1}\left[\|\mathbf{x}\| \leq X\right]$ where $X$ is a constant to be determined. Then for $\tilde{\mathbf{x}}$ and hypothesis class $\{\mathbf{w} : \|\mathbf{w}\| \leq W\}$, the loss $\ell(y\tilde{\mathbf{x}}^\top\mathbf{w})$ is bounded by $|y\tilde{\mathbf{x}}^\top\mathbf{w}| \leq WX$. By Corollary 4 in (Kakade et al., 2008), the following holds with probability at least $1 - \delta$: for all $\mathbf{w}$ such that $\|\mathbf{w}\| \leq W$,

$$\left|\mathbb{E}\ell(y\tilde{\mathbf{x}}^\top\mathbf{w}) - \frac{1}{n}\sum_{i=1}^{n}\ell(y_i\tilde{\mathbf{x}}_i^\top\mathbf{w})\right| \leq 2XW\sqrt{\frac{1}{n}} + XW\sqrt{\frac{\ln(1/(2\delta))}{2n}}.$$

For

$$X^2 = C\big(1 + \operatorname{tr}(\mathbf{\Sigma})\big)\ln(n/\delta),$$

by Lemma B.9, we have

$$\sup_{\|\mathbf{w}\|\leq W}\big|L(\mathbf{w}) - \mathbb{E}\ell(y\tilde{\mathbf{x}}^\top\mathbf{w})\big| \leq \frac{W}{n}.$$

Moreover, we have with probability $1 - \delta$, $\mathbf{x}_i = \tilde{\mathbf{x}}_i$ for $i = 1, \ldots, n$, which implies

$$\widehat{\mathcal{L}}(\mathbf{w}) = \frac{1}{n}\sum_{i=1}^{n}\ell(y_i\tilde{\mathbf{x}}_i^\top\mathbf{w}).$$

Putting things together with union bound, with probability at least $1 - \delta$, we have

$$\sup_{\|\mathbf{w}\|\leq W}\big(L(\mathbf{w}) - \widehat{\mathcal{L}}(\mathbf{w})\big) \leq \frac{W}{n} + 2XW\sqrt{\frac{1}{n}} + XW\sqrt{\frac{\ln(1/(3\delta))}{2n}}$$

$$\leq C_1 W\sqrt{\frac{\big(1 + \operatorname{tr}(\mathbf{\Sigma})\big)\ln(n/\delta)\ln(1/\delta)}{n}},$$

where $C_1 > 1$ is a constant. $\qquad\square$

*Proof of Theorem 3.1.* By the stopping condition and Lemma B.1, with probability at least $1 - \delta$, we have

$$\widehat{\mathcal{L}}(\mathbf{w}_t) \leq \widehat{\mathcal{L}}(\mathbf{w}_{0:k}^*), \quad \|\mathbf{w}_t - \mathbf{w}_{0:k}^*\| \leq 1 + \|\mathbf{w}_{0:k}^*\|.$$

Let $W := 1 + 2\|\mathbf{w}_{0:k}^*\|$. Then by Lemmas B.2 and B.3, with probability at least $1 - \delta$, we have

$$\begin{aligned}
&\mathcal{L}(\mathbf{w}_t) - \mathcal{L}(\mathbf{w}^*) \\
&= \mathcal{L}(\mathbf{w}_t) - \widehat{\mathcal{L}}(\mathbf{w}_t) + \widehat{\mathcal{L}}(\mathbf{w}_t) - \widehat{\mathcal{L}}(\mathbf{w}_{0:k}^*) + \widehat{\mathcal{L}}(\mathbf{w}_{0:k}^*) - \mathcal{L}(\mathbf{w}_{0:k}^*) + \mathcal{L}(\mathbf{w}_{0:k}^*) - \mathcal{L}(\mathbf{w}^*) \\
&\leq C_1 W\sqrt{\frac{\big(1 + \operatorname{tr}(\mathbf{\Sigma})\big)\ln(n/\delta)\ln(1/\delta)}{n}} + 0 + C_1 W\sqrt{\frac{\big(1 + \operatorname{tr}(\mathbf{\Sigma})\big)\ln(n/\delta)\ln(1/\delta)}{n}} + \frac{1}{2}\|\mathbf{w}_{k:\infty}^*\|_{\mathbf{\Sigma}}^2 \\
&\leq C(1 + \|\mathbf{w}_{0:k}^*\|)\sqrt{\frac{\big(1 + \operatorname{tr}(\mathbf{\Sigma})\big)\ln(n/\delta)\ln(1/\delta)}{n}} + \frac{1}{2}\|\mathbf{w}_{k:\infty}^*\|_{\mathbf{\Sigma}}^2.
\end{aligned}$$

This completes the proof. $\qquad\square$

### B.2. Proof of Theorem 3.2

Suppose that Assumption 1 holds throughout this subsection. In this subsection, we define

$$\begin{aligned}
W &:= 1 + 2\|\mathbf{w}^*\|, \\
X^2 &:= 2\operatorname{tr}(\mathbf{\Sigma}) + 2C(1 + \lambda_1)\ln\left(4C_1\big(1 + \lambda_1^{3/2}W^3\big)n\sqrt{\operatorname{tr}(\mathbf{\Sigma})}/\delta\right) \\
L &:= WX,
\end{aligned}$$

$$B := \frac{4C_1(1 + \lambda_1^{3/2}W^3)}{W^2X^2} \lesssim \sqrt{\lambda_1}W,$$

where $C_0, C_1 > 1$ are two sufficiently large constants. We aim to use tools from (Bartlett et al., 2005). To this end, consider the following random variables, function class, and loss function:

$$\tilde{\mathbf{x}} := \mathbf{x}\mathbb{1}\left[\|\mathbf{x}\| \leq X\right], \quad \mathcal{F} := \left\{\tilde{\mathbf{x}} \mapsto \frac{\mathbf{w}^\top \tilde{\mathbf{x}}}{WX} : \|\mathbf{w}\| \leq W\right\}, \quad \tilde{\ell} : t \mapsto \ell(WXt).$$

It is clear that $\|\tilde{\mathbf{x}}\| \leq X$ and $f(\tilde{\mathbf{x}}) \in [-1, 1]$ for every $f \in \mathcal{F}$. Recall that Rademacher complexity is defined as

$$\mathcal{R}_n\mathcal{F} := \sup_{f \in \mathcal{F}} \frac{1}{n} \sum_{i=1}^{n} \sigma_i f(\tilde{\mathbf{x}}_i),$$

where $\sigma_i$'s are independent Rademacher random variables. The following lemma is Corollary 5.3 in (Bartlett et al., 2005) restated in our context.

**Lemma B.4** (Corollary 5.3 in (Bartlett et al., 2005)). *Let $\mathcal{F}$ be a class of functions with ranges in $[-1, 1]$ and let $\tilde{\ell}$ be a loss function satisfying*

1. *There exists $f^* \in \mathcal{F}$ such that $\mathbb{E}\tilde{\ell}\big(yf^*(\tilde{\mathbf{x}})\big) = \inf_{f \in \mathcal{F}} \mathbb{E}\tilde{\ell}\big(yf(\tilde{\mathbf{x}})\big)$.*

2. *There exists $L$ such that $\tilde{\ell}$ is $L$-Lipschitz.*

3. *There exists $B \geq 1$ such that for every $f \in \mathcal{F}$,*

$$\mathbb{E}\big(f(\tilde{\mathbf{x}}) - f^*(\tilde{\mathbf{x}})\big)^2 \leq B\Big(\mathbb{E}\tilde{\ell}\big(yf(\tilde{\mathbf{x}})\big) - \mathbb{E}\tilde{\ell}\big(yf^*(\tilde{\mathbf{x}})\big)\Big).$$

*Let $\hat{f} \in \mathcal{F}$ be such that*

$$\frac{1}{n}\sum_{i=1}^{n} \tilde{\ell}(y_i\hat{f}(\tilde{\mathbf{x}}_i)) \leq \frac{1}{n}\sum_{i=1}^{n} \tilde{\ell}(y_if^*(\tilde{\mathbf{x}}_i)).$$

*Let $\phi$ be a sub-root function for which*

$$\psi(r) \geq BL\mathbb{E}\mathcal{R}_n\{f \in \mathcal{F} : L^2\mathbb{E}\big(f(\tilde{\mathbf{x}}) - f^*(\tilde{\mathbf{x}})\big)^2 \leq r\}.$$

*Then for any $r \geq \psi(r)$, with probability at least $1 - \delta$,*

$$\mathbb{E}\tilde{\ell}\big(y\hat{f}(\tilde{x})\big) - \mathbb{E}\tilde{\ell}\big(yf^*(\tilde{x})\big) \leq 705\frac{r}{B} + \frac{(11L + 27B)\ln(1/\delta)}{n}.$$

The following lemma provides a classical upper bound on the Rademacher complexity for the linear function class.

**Lemma B.5** (Theorem 6.5 in (Bartlett et al., 2005)). *We have*

$$\mathbb{E}\mathcal{R}_n\{f \in \mathcal{F} : L^2\mathbb{E}(f(\tilde{\mathbf{x}}) - f^*(\tilde{\mathbf{x}}))^2 \leq r\} \leq \sqrt{\frac{2k}{nL^2} \cdot r + \frac{\sum_{i>k} \lambda_i}{nX^2}},$$

*where $k$ is an arbitrary index.*

*Proof of Lemma B.5.* The proof is an adaptation of the proof of Theorem 6.5 in (Bartlett et al., 2005) in our context. We include it here for completeness. Assume $\mathbf{\Sigma}$ is diagonal without loss of generality. Let

$$\tilde{\mathbf{\Sigma}} := \max\left\{\frac{1}{W^2}\mathbf{I}, \ \frac{L^2}{2rW^2X^2}\mathbf{\Sigma}\right\},$$

where $\max\{\cdot, \cdot\}$ is applied entry wise. Recall that $0.5\mathbf{\Sigma} \preceq \mathbb{E}\tilde{\mathbf{x}}\tilde{\mathbf{x}}^\top \preceq \mathbf{\Sigma}$ by Lemma B.6. Then by definition, we have

$$\mathbb{E}\mathcal{R}_n\{f \in \mathcal{F} : L^2\mathbb{E}(f(\tilde{\mathbf{x}}) - f^*(\tilde{\mathbf{x}}))^2 \leq r\}$$

$$\leq \mathbb{E} \sup_{\|\mathbf{w}\| \leq W, \frac{L^2}{W^2 X^2} \|\mathbf{w} - \mathbf{w}^*\|_{\tilde{\boldsymbol{\Sigma}}}^2 \leq 2r} \left\langle \frac{1}{n} \sum_{i=1}^{n} \sigma_i \tilde{\mathbf{x}}_i, \frac{\mathbf{w}}{WX} \right\rangle = \frac{1}{WX} \mathbb{E} \sup_{\|\mathbf{w}\|_{\tilde{\boldsymbol{\Sigma}}} \leq 1} \left\langle \frac{1}{n} \sum_{i=1}^{n} \sigma_i \tilde{\mathbf{x}}_i, \mathbf{w} \right\rangle$$

$$\leq \frac{1}{WX} \mathbb{E} \left\| \frac{1}{n} \sum_{i=1}^{n} \sigma_i \tilde{\mathbf{x}}_i \right\|_{\tilde{\boldsymbol{\Sigma}}^{-1}} \leq \frac{1}{WX} \sqrt{\mathbb{E} \left\| \frac{1}{n} \sum_{i=1}^{n} \sigma_i \tilde{\mathbf{x}}_i \right\|_{\tilde{\boldsymbol{\Sigma}}^{-1}}^2} = \frac{1}{WX} \sqrt{\frac{1}{n} \mathbb{E} \|\tilde{\mathbf{x}}\|_{\tilde{\boldsymbol{\Sigma}}^{-1}}^2}$$

$$\leq \frac{1}{WX} \sqrt{\frac{1}{n} \langle \boldsymbol{\Sigma}, \tilde{\boldsymbol{\Sigma}}^{-1} \rangle} = \frac{1}{WX} \sqrt{\frac{1}{n} \sum_i \min \left\{ \frac{2rW^2 X^2}{L^2}, W^2 \lambda_i \right\}} \leq \sqrt{\frac{2k}{nL^2} \cdot r + \frac{\sum_{i>k} \lambda_i}{nX^2}},$$

where $k$ is an arbitrary index. This completes the proof. $\qquad \square$

The following lemma establishes several basic effects of clipping the random variable $\mathbf{x}$.

**Lemma B.6.** *There exists constant $C > 1$ such that for every $X^2 \geq 2\operatorname{tr}(\boldsymbol{\Sigma}) + C(1 + \lambda_1)t$, we have*

$$\Pr\left(\|\mathbf{x}\| \geq X\right) \leq \exp(-t), \quad t > 1.$$

*In addition, for every $C_1 > 1$, by setting $t \geq 2\ln(4C_1(1 + \lambda_1^{3/2} W^3))$ we have*

$$\mathbb{E}\mathbf{x}\mathbf{x}^\top \mathbb{1}\left[\|\mathbf{x}\| > X\right] \preceq \frac{1}{2C_1(1 + \lambda_1^{3/2} W^3)} \boldsymbol{\Sigma} \preceq \frac{1}{2} \boldsymbol{\Sigma}.$$

*In particular, this implies $\mathbb{E}\tilde{\mathbf{x}}\tilde{\mathbf{x}}^\top = \boldsymbol{\Sigma} - \mathbb{E}\mathbf{x}\mathbf{x}^\top \mathbb{1}\left[\|\mathbf{x}\| > X\right] \geq 0.5\boldsymbol{\Sigma}$.*

*Proof of Lemma B.6.* By Bernstein's inequality, there is a constant $c > 0$ such that

$$\Pr\left(\|\mathbf{x}\|^2 - \operatorname{tr}(\boldsymbol{\Sigma}) > z\right) \leq \exp\left(-c\min\left\{\frac{z^2}{\operatorname{tr}(\boldsymbol{\Sigma})}, \frac{z}{\lambda_1}\right\}\right).$$

We then obtain the first claim by setting $z = X^2 - \operatorname{tr}(\boldsymbol{\Sigma})$ and adjusting the constant.

To prove the second claim, for any unit vector $\mathbf{u}$, we have

$$\mathbb{E}(\mathbf{x}^\top \mathbf{u})^2 \mathbb{1}\left[\|\mathbf{x}\| \geq X\right] \leq \sqrt{\mathbb{E}(\mathbf{x}^\top \mathbf{u})^4} \cdot \sqrt{\mathbb{E}\mathbb{1}\left[\|\mathbf{x}\| \geq X\right]}$$
$$\leq \sqrt{3}\mathbb{E}(\mathbf{x}^\top \mathbf{u})^2 \cdot \sqrt{\Pr(\|\mathbf{x}\| \geq X)}$$
$$= \mathbf{u}^\top \boldsymbol{\Sigma} \mathbf{u} \cdot \sqrt{3}\exp(-t/2).$$

The above implies

$$\mathbb{E}\mathbf{x}\mathbf{x}^\top \mathbb{1}\left[\|\mathbf{x}\| \geq X\right] \preceq \sqrt{3}\exp(-t/2)\boldsymbol{\Sigma}.$$

Setting $t \geq 2\ln(4C_1(1 + \lambda_1 W^3))$ completes the proof. $\qquad \square$

The following lemma is from (Chardon et al., 2024).

**Lemma B.7.** *There exists a constant $C_1 > 0$ such that for every $\mathbf{w}$, we have*

$$\mathbb{E} \frac{1}{1 + \exp(\mathbf{x}^\top \mathbf{w})} \frac{1}{1 + \exp(-\mathbf{x}^\top \mathbf{w})} \mathbf{x}\mathbf{x}^\top \succeq \frac{1}{C_1(1 + \|\mathbf{w}\|_{\boldsymbol{\Sigma}}^3)} \boldsymbol{\Sigma}.$$

*Proof of Lemma B.7.* Let $s(t) := 1/(1 + e^{-t})$. Define

$$\beta := \|\mathbf{w}\|_{\boldsymbol{\Sigma}}, \quad \mathbf{u}_1 := \frac{1}{\beta} \boldsymbol{\Sigma}^{1/2} \mathbf{w}, \quad \mathbf{z} := \boldsymbol{\Sigma}^{-1/2} \mathbf{x} \sim \mathcal{N}(0, 1).$$

Then we only need to verify that

$$\mathbb{E}s'(\beta\mathbf{z}^\top\mathbf{u}_1)\mathbf{z}\mathbf{z}^\top \geq \frac{1}{C(1+\beta^3)}.$$

Hitting the left-hand side of the above with $\mathbf{u}_1$, we get

$$\mathbb{E}s'(\beta\mathbf{z}^\top\mathbf{u}_1)(\mathbf{z}^\top\mathbf{u}_1)^2 = \mathbb{E}_{g\sim\mathcal{N}(0,1)}s'(\beta g)g^2 \geq \sqrt{\frac{2}{\pi}}\frac{2^3}{3}\min\left\{\frac{1}{4e^4\beta^3}, \frac{s'(2)}{e^2}\right\} \geq \frac{1}{C_1(1+\beta^3)},$$

where the first inequality is by Lemma 25 in Chardon et al. (2024). Hitting that again with a unit vector $\mathbf{u}_2$ such that $\mathbf{u}_1^\top\mathbf{u}_2 = 0$, we get

$$\mathbb{E}s'(\beta\mathbf{z}^\top\mathbf{u}_1)(\mathbf{z}^\top\mathbf{u}_2)^2 = \mathbb{E}_{g\sim\mathcal{N}(0,1)}s'(\beta g) \geq \sqrt{\frac{2}{\pi}}\frac{2^2}{2}\min\left\{\frac{1}{4e^4\beta}, \frac{s'(2)}{e^2}\right\} \geq \frac{1}{C_1(0.5+\beta)} \geq \frac{1}{C_1(1+\beta^3)},$$

where the first inequality is again by Lemma 25 in Chardon et al. (2024). Together, these two lower bounds and the properties of Gaussian complete the proof. $\qquad\square$

The following lemma verifies a key condition in Lemma B.4.

**Lemma B.8.** *Let $X^2 \geq 2\operatorname{tr}(\boldsymbol{\Sigma}) + 2C(1+\lambda_1)\ln\left(4C_1(1+\lambda_1^{3/2}W^3)\right)$ and $B \geq 4C_1(1+\lambda_1^{3/2}W^3)/(W^2X^2)$. Then for every $\|\mathbf{w}\| \leq W$, we have*

$$\mathbb{E}\left(\tilde{\mathbf{x}}^\top\mathbf{w} - \tilde{\mathbf{x}}^\top\mathbf{w}^*\right)^2 \leq W^2X^2B\left(\mathbb{E}\ell(y\tilde{\mathbf{x}}^\top\mathbf{w}) - \mathbb{E}\ell(y\tilde{\mathbf{x}}^\top\mathbf{w}^*)\right).$$

*Proof of Lemma B.8.* Since $\mathbb{E}\tilde{\mathbf{x}}\tilde{\mathbf{x}}^\top \preceq \boldsymbol{\Sigma}$, it suffices to show that

$$\|\mathbf{w} - \mathbf{w}^*\|_{\boldsymbol{\Sigma}}^2 \leq W^2X^2B\left(\mathbb{E}\ell(y\tilde{\mathbf{x}}^\top\mathbf{w}) - \mathbb{E}\ell(y\tilde{\mathbf{x}}^\top\mathbf{w}^*)\right).$$

Recall that $\mathbf{w}^*$ is the minimizer of $\mathbb{E}_y\ell(y\tilde{\mathbf{x}}^\top\mathbf{w})$ (where the expectation is conditional on $\tilde{\mathbf{x}}$, see the proof of Proposition 2.1 in Appendix A.1). By the midpoint theorem, there exists $\mathbf{v}$ between $\mathbf{w}$ and $\mathbf{w}^*$ (thus $\|\mathbf{v}\| \leq W$) such that

$$\mathbb{E}\ell(y\tilde{\mathbf{x}}^\top\mathbf{w}) - \mathbb{E}\ell(y\tilde{\mathbf{x}}^\top\mathbf{w}^*) = \frac{1}{2}\mathbb{E}\ell''(y\tilde{\mathbf{x}}^\top\mathbf{v})\langle\tilde{\mathbf{x}}^{\otimes 2}, (\mathbf{w}-\mathbf{w}^*)^{\otimes 2}\rangle.$$

Thus it suffices to show

$$\text{for every } \mathbf{v} \text{ such that } \|\mathbf{v}\| \leq W, \quad \mathbb{E}\ell''(y\tilde{\mathbf{x}}^\top\mathbf{v})\tilde{\mathbf{x}}\tilde{\mathbf{x}}^\top \succeq \frac{2}{W^2X^2B}\boldsymbol{\Sigma}.$$

This is because

$$\begin{aligned}
\mathbb{E}\ell''(y\tilde{\mathbf{x}}^\top\mathbf{v})\tilde{\mathbf{x}}\tilde{\mathbf{x}}^\top &= \mathbb{E}\frac{1}{1+\exp(\tilde{\mathbf{x}}^\top\mathbf{v})}\frac{1}{1+\exp(-\tilde{\mathbf{x}}^\top\mathbf{v})}\tilde{\mathbf{x}}\tilde{\mathbf{x}}^\top \\
&= \mathbb{E}\frac{1}{1+\exp(\mathbf{x}^\top\mathbf{v})}\frac{1}{1+\exp(-\mathbf{x}^\top\mathbf{v})}\mathbf{x}\mathbf{x}^\top\mathbb{1}\left[\|\mathbf{x}\| \leq X\right] \\
&\succeq \mathbb{E}\frac{1}{1+\exp(\mathbf{x}^\top\mathbf{v})}\frac{1}{1+\exp(-\mathbf{x}^\top\mathbf{v})}\mathbf{x}\mathbf{x}^\top - \mathbb{E}\mathbf{x}\mathbf{x}^\top\mathbb{1}\left[\|\mathbf{x}\| > X\right].
\end{aligned}$$

By Lemma B.7, we have

$$\mathbb{E}\frac{1}{1+\exp(\mathbf{x}^\top\mathbf{v})}\frac{1}{1+\exp(-\mathbf{x}^\top\mathbf{v})}\mathbf{x}\mathbf{x}^\top \succeq \frac{1}{C_1(1+\|\mathbf{v}\|_{\boldsymbol{\Sigma}}^3)}\boldsymbol{\Sigma} \succeq \frac{1}{C_1(1+\lambda_1^{3/2}W^3)}\boldsymbol{\Sigma},$$

where $C_1 > 1$ is a constant. By Lemma B.6, we have

$$\mathbb{E}\mathbf{x}\mathbf{x}^\top\mathbb{1}\left[\|\mathbf{x}\| > X\right] \preceq \frac{1}{2C_1(1+\lambda_1^{3/2}W^3)}\boldsymbol{\Sigma}.$$

So we have

$$\mathbb{E}\ell''(y\tilde{\mathbf{x}}^\top\mathbf{v})\tilde{\mathbf{x}}\tilde{\mathbf{x}}^\top \succeq \frac{1}{2C_1(1+\lambda_1^{3/2}W^3)}\mathbf{\Sigma}.$$

We complete the proof by noting that $W^2X^2B \geq 4C_1(1+\lambda_1^{3/2}W^3)$. □

The following lemma controls the effect of clipping on population risk.

**Lemma B.9.** *Let* $X^2 \geq 2\operatorname{tr}(\mathbf{\Sigma}) + C(1+\lambda_1)\cdot 2\ln\left(n\sqrt{\operatorname{tr}(\mathbf{\Sigma})}\right)$. *Then for every* $\mathbf{w}$ *such that* $\|\mathbf{w}\| \leq W$, *we have*

$$\left|L(\mathbf{w}) - \mathbb{E}\ell(y\tilde{\mathbf{x}}^\top\mathbf{w})\right| \leq \frac{W}{n}.$$

*Proof of Lemma B.9.* By definition and 1-Lipschitzness of $\ell$, we have

$$\begin{aligned}
\left|L(\mathbf{w}) - \mathbb{E}\ell(y\tilde{\mathbf{x}}^\top\mathbf{w})\right| &= \left|\mathbb{E}\ell(y\mathbf{x}^\top\mathbf{w}) - \ell\left(y\mathbf{x}^\top\mathbf{w}\mathbb{1}\left[\|\mathbf{x}\| \leq X\right]\right)\right| \\
&\leq \mathbb{E}\left|y\mathbf{x}^\top\mathbf{w}\mathbb{1}\left[\|\mathbf{x}\| > X\right]\right| \\
&\leq W\mathbb{E}\|\mathbf{x}\|\mathbb{1}\left[\|\mathbf{x}\| > X\right] \\
&\leq W\sqrt{\mathbb{E}\|\mathbf{x}\|^2}\sqrt{\Pr(\|\mathbf{x}\| > X)} \\
&\leq W\sqrt{\operatorname{tr}(\mathbf{\Sigma})}\exp(-t/2) \leq \frac{W}{n},
\end{aligned}$$

where we use Lemma B.6 and the choice of $X$. This completes the proof. □

The following lemma shows the early-stopped GD has a small norm.

**Lemma B.10.** *Let* $\beta := C_0\left(1 + \operatorname{tr}(\mathbf{\Sigma}) + \lambda_1\ln(1/\delta)/n\right)$, *where* $C_0 > 1$ *is a sufficiently large constant. Assume that* $\eta \leq 1/\beta$ *and* $t$ *is such that* $\widehat{\mathcal{L}}(\mathbf{w}^*) \leq \widehat{\mathcal{L}}(\mathbf{w}_{t-1})$. *Then with probability at least* $1-\delta$, *we have* $\|\mathbf{w}_t - \mathbf{w}^*\| \leq 1 + \|\mathbf{w}\|$.

*Proof of Lemma B.10.* This is the same as the proof of Lemma B.1. □

We are now ready to proof Theorem 3.2 using Lemma B.4.

*Proof of Theorem 3.2.* It is clear that $\tilde{\ell}$ is $L$-Lipschitz and that $f^* := \langle\mathbf{w}^*, \cdot\rangle/(WX)$ satisfies

$$f^* \in \arg\inf_{f\in\mathcal{F}} \mathbb{E}\tilde{\ell}\left(yf(\tilde{\mathbf{x}})\right).$$

Moreover, for every $f = \langle\mathbf{w}, \cdot\rangle/(WX) \in \mathcal{F}$, by Lemma B.8, we have

$$\begin{aligned}
\mathbb{E}\left(f(\tilde{\mathbf{x}}) - f^*(\tilde{\mathbf{x}})\right)^2 &= \frac{1}{W^2X^2}\mathbb{E}(y\mathbf{x}^\top\mathbf{w} - y\mathbf{x}^\top\mathbf{w}^*)^2 \\
&\leq B\left(\mathbb{E}\ell(y\tilde{\mathbf{x}}^\top\mathbf{w}) - \mathbb{E}\ell(y\tilde{\mathbf{x}}^\top\mathbf{w}^*)\right) = B\left(\mathbb{E}\tilde{\ell}\left(yf(\tilde{\mathbf{x}})\right) - \mathbb{E}\tilde{\ell}\left(yf^*(\tilde{\mathbf{x}})\right)\right).
\end{aligned}$$

So far, we have verified the conditions on $\mathcal{F}$ and $\tilde{\ell}$ for applying Lemma B.4.

Consider the function

$$\hat{f} := \langle\mathbf{w}_t, \cdot\rangle/(WX).$$

By Lemma B.10 we have $\|\mathbf{w}_t\| \leq W$, thus $\hat{f} \in \mathcal{F}$. Moreover, by the choice of $X$ and Lemma B.6, with probability $1-\delta$, we have $\mathbf{x}_i = \tilde{\mathbf{x}}_i$ for all $i = 1, \ldots, n$. Thus the stopping criterion implies

$$\frac{1}{n}\sum_{i=1}^n \tilde{\ell}\left(y_i\hat{f}(\tilde{\mathbf{x}}_i)\right) = \frac{1}{n}\sum_{i=1}^n \ell\left(y_i\mathbf{x}_i^\top\mathbf{w}_t\right) = \widehat{\mathcal{L}}(\mathbf{w}_t) \leq \widehat{\mathcal{L}}(\mathbf{w}^*) = \frac{1}{n}\sum_{i=1}^n \tilde{\ell}\left(y_if^*(\tilde{\mathbf{x}}_i)\right).$$

So we can apply Lemma B.4 to $\hat{f}$. With probability at least $1 - \delta$, we have

$$\mathbb{E}\tilde{\ell}\big(y\hat{f}(\tilde{\mathbf{x}})\big) - \mathbb{E}\tilde{\ell}\big(yf^*(\tilde{\mathbf{x}})\big) \leq 705\frac{r}{B} + \frac{(11L + 27B)\ln(1/\delta)}{n},$$

where, by Lemmas B.4 and B.5, $r$ is such that

$$r \geq \psi(r) := BL\sqrt{\frac{2k}{nL^2} \cdot r + \frac{\sum_{i>k}\lambda_i}{nX^2}}.$$

Choosing

$$r = \frac{4kB^2}{n} + \sqrt{\frac{2B^2L^2\sum_{i>k}\lambda_i}{nX^2}}$$

we have

$$\mathbb{E}\tilde{\ell}\big(y\hat{f}(\tilde{\mathbf{x}})\big) - \mathbb{E}\tilde{\ell}\big(yf^*(\tilde{\mathbf{x}})\big) \lesssim \frac{r}{B} + \frac{(L+B)\ln(1/\delta)}{n}$$

$$\lesssim \frac{Bk}{n} + \frac{L}{X}\sqrt{\frac{\sum_{i>k}\lambda_i}{n}} + \frac{(L+B)\ln(1/\delta)}{n}.$$

Then applying Lemma B.9, we get

$$L(\mathbf{w}_t) - L(\mathbf{w}^*) \leq \mathbb{E}\tilde{\ell}\big(y\hat{f}(\tilde{\mathbf{x}})\big) - \mathbb{E}\tilde{\ell}\big(yf^*(\tilde{\mathbf{x}})\big) + \frac{2W}{n}$$

$$\lesssim \frac{Bk}{n} + \frac{L}{X}\sqrt{\frac{\sum_{i>k}\lambda_i}{n}} + \frac{(L+B)\ln(1/\delta)}{n} + \frac{2W}{n}$$

$$\lesssim \max\{\|\mathbf{w}^*\|, 1\}\left(\frac{k\lambda_1^{1/2}}{n} + \sqrt{\frac{\sum_{i>k}\lambda_i}{n}} + \frac{\ln(1/\delta)\sqrt{(1 + \operatorname{tr}(\mathbf{\Sigma}))\ln\big((1 + \operatorname{tr}(\mathbf{\Sigma})\|\mathbf{w}^*\|)n/\delta\big)}}{n}\right)$$

$$\lesssim \|\mathbf{w}^*\|\left(\frac{k}{n} + \sqrt{\frac{\sum_{i>k}\lambda_i}{n}} + \frac{\ln(1/\delta)\sqrt{\operatorname{tr}(\mathbf{\Sigma})\ln\big(n\|\mathbf{w}^*\|\operatorname{tr}(\mathbf{\Sigma})/\delta\big)}}{n}\right),$$

where we assume $\|\mathbf{w}^*\| \gtrsim 1$, $\lambda_1 \lesssim 1$, and $\operatorname{tr}(\mathbf{\Sigma}) \gtrsim 1$. This completes the proof. $\qquad\square$

## C. Lower Bounds for Interpolating Estimators

### C.1. Proof of Theorem 4.1

*Proof of Theorem 4.1.* Consider a sequence of estimators $(\mathbf{w}_t)_{t\geq 0}$ such that

$$\|\mathbf{w}_t\| \to \infty, \quad \mathbf{w}_t/\|\mathbf{w}_t\| \to \tilde{\mathbf{w}},$$

where $\tilde{\mathbf{w}}$ is a fixed unit vector. Fix a small constant $\gamma > 0$. Define an event

$$\mathcal{F} := \{\mathbf{x} : |\mathbf{x}^\top\mathbf{w}^*| \leq 10\|\mathbf{w}^*\|_{\mathbf{\Sigma}}, \ \|\mathbf{x}\| \leq 10\sqrt{\operatorname{tr}(\mathbf{\Sigma})}, \ |\mathbf{x}^\top\tilde{\mathbf{w}}| \geq \gamma\}.$$

Since $\mathbf{x} \sim \mathcal{N}(0, \mathbf{\Sigma})$ and $\|\tilde{\mathbf{w}}\|_{\mathbf{\Sigma}} > 0$, we have $\Pr(\mathcal{F}) > 0$. Let $t_0$ be such that

$$\left\|\frac{\mathbf{w}_t}{\|\mathbf{w}_t\|} - \tilde{\mathbf{w}}\right\| \leq \frac{\gamma}{20\sqrt{\operatorname{tr}(\mathbf{\Sigma})}}, \quad \text{for every } t \geq t_0.$$

Then for every $\mathbf{x} \in \mathcal{F}$ and $t \geq t_0$, we have

$$\frac{|\mathbf{x}^\top\mathbf{w}_t|}{\|\mathbf{w}_t\|} \geq |\mathbf{x}^\top\tilde{\mathbf{w}}| - \|\mathbf{x}\| \cdot \left\|\frac{\mathbf{w}_t}{\|\mathbf{w}_t\|} - \tilde{\mathbf{w}}\right\| \geq \gamma - 10\sqrt{\operatorname{tr}(\mathbf{\Sigma})} \cdot \frac{\gamma}{20\sqrt{\operatorname{tr}(\mathbf{\Sigma})}} \geq \frac{\gamma}{2}.$$

Then the population risk of $\mathbf{w}_t$ is

$$
\begin{aligned}
\mathcal{L}(\mathbf{w}_t) &= \mathbb{E}_{\mathbf{x}}\mathbb{E}_y \ln(1 + \exp(-y\mathbf{x}^\top \mathbf{w}_t)) \\
&= \mathbb{E} \sum_{y\in\{\pm 1\}} \frac{1}{1 + \exp(-y\mathbf{x}^\top \mathbf{w}^*)} \ln(1 + \exp(-y\mathbf{x}^\top \mathbf{w}_t)) \\
&\geq \mathbb{E} \sum_{y\in\{\pm 1\}} \frac{1}{1 + \exp(-y\mathbf{x}^\top \mathbf{w}^*)} \ln(1 + \exp(-y\mathbf{x}^\top \mathbf{w}_t))\mathbb{1}\left[\mathbf{x} \in \mathcal{F}\right] \\
&\geq \mathbb{E} \frac{\ln(1 + \exp(|\mathbf{x}^\top \mathbf{w}_t|))}{1 + \exp(|\mathbf{x}^\top \mathbf{w}^*|)}\mathbb{1}\left[\mathbf{x} \in \mathcal{F}\right] \\
&\geq \mathbb{E} \frac{\|\mathbf{w}_t\|\gamma/2}{1 + \exp(10\|\mathbf{w}^*\|_{\boldsymbol{\Sigma}})}\mathbb{1}\left[\mathbf{x} \in \mathcal{F}\right] \\
&\geq \frac{\|\mathbf{w}_t\|\gamma/2}{1 + \exp(10\|\mathbf{w}^*\|)}\Pr(\mathcal{F}) \to \infty,
\end{aligned}
$$

where the last inequality is because $\|\mathbf{w}_t\| \to \infty$.

Now we consider the calibration error. Under event $\mathcal{F}$, we have

$$
\begin{aligned}
p(\mathbf{w}^*; \mathbf{x}) = \frac{1}{1 + \exp(-\mathbf{x}^\top \mathbf{w}^*)} &\in \left[\frac{1}{1 + \exp(10\|\mathbf{w}^*\|_{\boldsymbol{\Sigma}})}, \frac{1}{1 + \exp(-10\|\mathbf{w}^*\|_{\boldsymbol{\Sigma}})}\right] \\
&\in \left[\exp(-10\|\mathbf{w}^*\|_{\boldsymbol{\Sigma}}),\ 1 - \exp(-10\|\mathbf{w}^*\|_{\boldsymbol{\Sigma}})\right].
\end{aligned}
$$

Moreover, under event $\mathcal{F}$, for $t > t_0$, we have

$$
p(\mathbf{w}_t; \mathbf{x}) = \frac{1}{1 + \exp(-\mathbf{x}^\top \mathbf{w}_t)}
\begin{cases}
\leq \dfrac{1}{1 + \exp(\gamma\|\mathbf{w}_t\|/2)} \leq \exp(-\gamma\|\mathbf{w}_t\|/2) & \mathbf{x}^\top \mathbf{w}_t < 0, \\[2mm]
\geq \dfrac{1}{1 + \exp(-\gamma\|\mathbf{w}_t\|/2)} \geq 1 - \exp(-\gamma\|\mathbf{w}_t\|/2) & \mathbf{x}^\top \mathbf{w}_t > 0.
\end{cases}
$$

These together imply that

$$
|p(\mathbf{w}_t; \mathbf{x}) - p(\mathbf{w}^*; \mathbf{x})| \geq \exp(-10\|\mathbf{w}^*\|_{\boldsymbol{\Sigma}}) - \exp(-\gamma\|\mathbf{w}_t\|/2) \to \exp(-10\|\mathbf{w}^*\|_{\boldsymbol{\Sigma}}).
$$

Then for the calibration error, we have

$$
\begin{aligned}
\mathcal{C}(\mathbf{w}_t) &= \mathbb{E}|p(\mathbf{w}_t; \mathbf{x}) - p(\mathbf{w}^*; \mathbf{x})|^2 \\
&\geq \mathbb{E}|p(\mathbf{w}_t; \mathbf{x}) - p(\mathbf{w}^*; \mathbf{x})|^2\mathbb{1}\left[\mathbf{x} \in \mathcal{F}\right] \\
&\geq \left(\exp(-10\|\mathbf{w}^*\|_{\boldsymbol{\Sigma}}) - \exp(-\gamma\|\mathbf{w}_t\|/2)\right)^2 \Pr(\mathcal{F}) \\
&\to \exp(-20\|\mathbf{w}^*\|_{\boldsymbol{\Sigma}})\Pr(\mathcal{F}) > 0.
\end{aligned}
$$

This completes the proof. $\qquad\square$

### C.2. Proof of Theorem 4.2

**Lemma C.1.** *For a non-zero vector* $\mathbf{w}$, *let* $\theta$ *be the angle between* $\boldsymbol{\Sigma}^{1/2}\mathbf{w}$ *and* $\boldsymbol{\Sigma}^{1/2}\mathbf{w}^*$. *Then*

$$
\mathcal{E}(\mathbf{w}) - \mathcal{E}(\mathbf{w}^*) \geq
\begin{cases}
\dfrac{1}{4\sqrt{2\pi}\|\mathbf{w}^*\|_{\boldsymbol{\Sigma}}} & \dfrac{\pi}{2} < \theta \leq \pi, \\[4mm]
\dfrac{\|\mathbf{w}^*\|_{\boldsymbol{\Sigma}}}{48\pi \max\{1, \|\mathbf{w}^*\|_{\boldsymbol{\Sigma}}^3\}}(1 - \cos(\theta)) & 0 \leq \theta \leq \dfrac{\pi}{2}.
\end{cases}
$$

*Proof of Lemma C.1.* Let $s(t) = 1/(1 + e^{-t})$. Notice that for $t > 0$,

$$
s(t) - 1/2 = \frac{1 - \exp(-t)}{2(1 + \exp(-t))} \geq \frac{1 - \exp(-t)}{4} \geq \frac{1 - 1/(t+1)}{4} \geq \frac{t}{8}\mathbb{1}\left[0 < t < 1\right].
$$

Since $\mathbf{x}$ and $-\mathbf{x}$ are identically distributed, we have

$$
\begin{aligned}
&\Pr(y\mathbf{x}^\top \mathbf{w} \leq 0) - \Pr(y\mathbf{x}^\top \mathbf{w}^* \leq 0) \\
&= 2\mathbb{E}\mathbb{1}\left[\mathbf{x}^\top \mathbf{w} < 0, \ \mathbf{x}^\top \mathbf{w}^* > 0\right] |s(\mathbf{x}^\top \mathbf{w}^*) - 1/2| \\
&= 2\mathbb{E}\mathbb{1}\left[\mathbf{x}^\top \mathbf{w} < 0, \ \mathbf{x}^\top \mathbf{w}^* > 0\right] \left(s(\mathbf{x}^\top \mathbf{w}^*) - 1/2\right) \\
&\geq \frac{1}{4}\mathbb{E}\mathbf{x}^\top \mathbf{w}^* \mathbb{1}\left[\mathbf{x}^\top \mathbf{w} < 0, \ 0 < \mathbf{x}^\top \mathbf{w}^* < 1\right].
\end{aligned}
$$

Without loss of generality, assume that $\|\mathbf{w}\|_{\mathbf{\Sigma}} = 1$. We can write $\mathbf{\Sigma}^{1/2}\mathbf{w} = \mathbf{\Sigma}^{1/2}\mathbf{w}^*/\|\mathbf{w}^*\|_{\mathbf{\Sigma}}\cos\theta - \mathbf{v}_\perp \sin\theta$, where $\mathbf{v}_\perp$ is a unit vector such that $\langle \mathbf{v}_\perp, \mathbf{\Sigma}^{1/2}\mathbf{w}^*\rangle = 0$. Since $\mathbf{x} \sim \mathcal{N}(0, \mathbf{\Sigma})$, we have

$$
\begin{aligned}
&\Pr(y\mathbf{x}^\top \mathbf{w} \leq 0) - \Pr(y\mathbf{x}^\top \mathbf{w}^* \leq 0) \\
&\geq \frac{\|\mathbf{w}^*\|_{\mathbf{\Sigma}}}{4}\mathbb{E}_{g_1,g_2\sim\mathcal{N}(0,1)}g_1\mathbb{1}\left[g_1\cos(\theta) - g_2\sin(\theta) < 0, \ 0 < g_1 < 1/\|\mathbf{w}^*\|_{\mathbf{\Sigma}}\right] \\
&= \frac{\|\mathbf{w}^*\|_{\mathbf{\Sigma}}}{8\pi}\underbrace{\int_{0<g_1<1/\|\mathbf{w}^*\|_{\mathbf{\Sigma}},\, g_1\cos\theta<g_2\sin\theta} g_1 \exp\left(-(g_1^2+g_2^2)/2\right)\mathrm{d}g_1\mathrm{d}g_2}_{\diamondsuit}.
\end{aligned}
$$

To proceed, we discuss two cases.

- If $\pi/2 < \theta \leq \pi$, we have $\cos\theta < 0$ and $\sin\theta \geq 0$. So we have

$$
\begin{aligned}
\diamondsuit &\geq \int_{0<g_1<1/\|\mathbf{w}^*\|_{\mathbf{\Sigma}},\, 0<g_2} g_1 \exp\left(-(g_1^2+g_2^2)/2\right)\mathrm{d}g_1\mathrm{d}g_2 \\
&= \frac{\sqrt{2\pi}}{2}\int_{0<g_1<1/\|\mathbf{w}^*\|_{\mathbf{\Sigma}}} g_1 \exp\left(-g_1^2/2\right)\mathrm{d}g_1 \\
&= \sqrt{2\pi}\left(1 - \exp(-1/\|\mathbf{w}^*\|_{\mathbf{\Sigma}}^2)\right) \\
&\geq \frac{\sqrt{2\pi}}{\|\mathbf{w}^*\|_{\mathbf{\Sigma}}^2}.
\end{aligned}
$$

  So we have

$$
\Pr(y\mathbf{x}^\top \mathbf{w} \leq 0) - \Pr(y\mathbf{x}^\top \mathbf{w}^* \leq 0) \geq \frac{\|\mathbf{w}^*\|_{\mathbf{\Sigma}}}{8\pi}\cdot\diamondsuit \geq \frac{1}{4\sqrt{2\pi}\|\mathbf{w}^*\|_{\mathbf{\Sigma}}}.
$$

- If $0 \leq \theta \leq \pi/2$, we have $\cos\theta \geq 0$. By changing of variables, we get

$$
\begin{aligned}
\diamondsuit &= \int_{0<g_1<1/\|\mathbf{w}^*\|_{\mathbf{\Sigma}},\, g_1<g_2\tan\theta} g_1 \exp\left(-(g_1^2+g_2^2)/2\right)\mathrm{d}g_1\mathrm{d}g_2 \\
&= \int_{0<r\sin\psi<1/\|\mathbf{w}^*\|_{\mathbf{\Sigma}},0<\psi<\theta} r\sin\psi \exp\left(-r^2/2\right)\cdot r\mathrm{d}r\mathrm{d}\psi \\
&\geq \int_{0<\psi<\theta} \sin\psi\left(\int_{0<r<1/\|\mathbf{w}^*\|_{\mathbf{\Sigma}}} r^2 \exp\left(-\frac{1}{2}r^2\right)\mathrm{d}r\right)\mathrm{d}\psi \\
&\geq \int_{0<\psi<\theta} \sin\psi\left(\int_{0<r<\min\{1,1/\|\mathbf{w}^*\|_{\mathbf{\Sigma}}\}} r^2\left(1 - \frac{1}{2}r^2\right)\mathrm{d}r\right)\mathrm{d}\psi \\
&\geq \int_{0<\psi<\theta} \sin\psi\left(\frac{1}{2}\int_{0<r<\min\{1,1/\|\mathbf{w}^*\|_{\mathbf{\Sigma}}\}} r^2\right)\mathrm{d}\psi \\
&= \frac{1}{6\max\{1,\|\mathbf{w}^*\|_{\mathbf{\Sigma}}^3\}}\int_{0<\psi<\theta} \sin\psi\mathrm{d}\psi \\
&= \frac{1}{6\max\{1,\|\mathbf{w}^*\|_{\mathbf{\Sigma}}^3\}}\left(1 - \cos\theta\right).
\end{aligned}
$$

So we have

$$\Pr(y\mathbf{x}^\top \mathbf{w} \le 0) - \Pr(y\mathbf{x}^\top \mathbf{w}^* \le 0) \ge \frac{\|\mathbf{w}^*\|_\mathbf{\Sigma}}{8\pi} \cdot \diamond \ge \frac{\|\mathbf{w}^*\|_\mathbf{\Sigma}}{48\pi \max\{1, \|\mathbf{w}^*\|_\mathbf{\Sigma}^3\}}(1 - \cos(\theta)).$$

This completes the proof. $\qquad\square$

**Lemma C.2.** *Let* $\mathbf{z} \sim \mathcal{N}(0, \mathbf{I})$ *and* $\Pr(y|\mathbf{z}) = s(y\mathbf{z}^\top \mathbf{v}^*)$. *Then for any unit vector* $\mathbf{v}$*, we have*

$$\Pr(y\mathbf{z}^\top \mathbf{v} < -0.5) \ge \frac{0.25}{1 + \exp(\|\mathbf{v}^*\|)}.$$

*Proof of Lemma C.2.* This is by direct calculation.

$$
\begin{aligned}
\mathbb{E}\mathbb{1}\left[y\mathbf{z}^\top \mathbf{v} < -0.5\right] &= \mathbb{E}\mathbb{1}\left[y = 1,\ \mathbf{z}^\top \mathbf{v} < -0.5\right] + \mathbb{1}\left[y = -1,\ \mathbf{z}^\top \mathbf{v} > 0.5\right] \\
&= \mathbb{E}s(\mathbf{z}^\top \mathbf{v}^*)\mathbb{1}\left[\mathbf{z}^\top \mathbf{v} < -0.5\right] + s(-\mathbf{z}^\top \mathbf{v}^*)\mathbb{1}\left[\mathbf{z}^\top \mathbf{v} > 0.5\right] \\
&\ge \mathbb{E}\frac{1}{1 + \exp(|\mathbf{z}^\top \mathbf{v}^*|)}\mathbb{1}\left[|\mathbf{z}^\top \mathbf{v}| > 0.5\right] \\
&\ge \mathbb{E}\frac{1}{1 + \exp(\|\mathbf{v}^*\|)}\mathbb{1}\left[|\mathbf{z}^\top \mathbf{v}| > 0.5,\ |\mathbf{z}^\top \mathbf{v}^*| < \|\mathbf{v}^*\|\right] \\
&\ge \frac{1}{1 + \exp(\|\mathbf{v}^*\|)}\left(1 - \Pr(|\mathbf{z}^\top \mathbf{v}| \le 0.5) - \Pr(|\mathbf{z}^\top \mathbf{v}^*| \ge \|\mathbf{v}^*\|)\right) \\
&= \frac{1}{1 + \exp(\|\mathbf{v}^*\|)}\left(1 - (\Phi(0.5) - \Phi(-0.5)) - 2(1 - \Phi(1))\right) \\
&\ge \frac{0.25}{1 + \exp(\|\mathbf{v}^*\|)},
\end{aligned}
$$

where $\Phi$ is the cumulative distribution function of normal distribution. We complete the proof. $\qquad\square$

**Lemma C.3.** *Let* $(y_i, \mathbf{z}_i)_{i=1}^n$ *be independent copies of* $(y, \mathbf{z})$. *Assume that*

$$n \ge C\left(1 + \exp(\|\mathbf{v}^*\|)\right)k\ln(k/\delta)$$

*for a sufficiently large constant* $C > 1$. *Let* $\mathcal{S} := \{\mathbf{v} : \|\mathbf{v}_{0:k}\| = 1, \mathbf{v}_{k:\infty} = 0\}$. *Then*

$$\Pr\left(\text{for every } \mathbf{v} \in \mathcal{S},\ \#\{i \in [n] : y_i\mathbf{z}_i^\top \mathbf{v} \le -0.4\} \ge \frac{n/8}{1 + \exp(\|\mathbf{v}^*\|)}\right) \ge 1 - \delta.$$

*Proof of Lemma C.3.* For each unit vector $\mathbf{v}$, define a binary random variable

$$\xi(\mathbf{v}) := \mathbb{1}\left[y\mathbf{z}^\top \mathbf{v} \le -0.5\right] \in \{0, 1\}.$$

Let $(\xi_i(\mathbf{v}))_{i=1}^n$ be independent copies of $\xi(\mathbf{v})$. By the multiplicative Chernoff bound, we have

$$\Pr\left(\sum_{i=1}^n \xi_i(\mathbf{v}) \le 0.5n\mathbb{E}\xi(\mathbf{v})\right) \le \exp\left(-n\mathbb{E}\xi(\mathbf{v})/8\right).$$

By Lemma C.2, we have $\mathbb{E}\xi(\mathbf{v}) \ge 0.25/\left(1 + \exp\|\mathbf{v}^*\|\right)$. Thus we have

$$\Pr\left(\sum_{i=1}^n \xi_i(\mathbf{v}) \le \frac{n/8}{1 + \exp\|\mathbf{v}^*\|}\right) \le \exp\left(-\frac{n/32}{1 + \exp(\|\mathbf{v}^*\|)}\right).$$

Let $\mathcal{C}$ be an $\epsilon$-$\ell_2$-covering of the $k$-dimensional unit sphere $\mathcal{S}$. Then $|\mathcal{C}| = \mathcal{O}((1/\epsilon)^k)$. By a union bound, we get

$$\Pr\left(\text{for every } \mathbf{v} \in \mathcal{C},\ \sum_{i=1}^n \xi_i(\mathbf{v}) \ge \frac{n/8}{1 + \exp(\|\mathbf{v}^*\|)}\right) \ge 1 - \exp\left(-\frac{n/32}{1 + \exp(\|\mathbf{v}^*\|)} + Ck\ln(1/\epsilon)\right).$$

Here $C > 1$ is a sufficiently large constant and may vary line by line. Moreover, with probability at least $1 - 0.5\delta$, we have

$$\Pr\left(\max_{i\in[n]} \|\mathbf{z}_{0:k}^{(i)}\| \leq C\sqrt{k\ln(n/\delta)}\right) \geq 1 - 0.5\delta.$$

Under the joint of the two events, for every $\mathbf{v} \in \mathcal{S}$, there is a $\mathbf{v}' \in \mathcal{C}$ such that

$$\text{for every } i \in [n], \ y_i\mathbf{z}_i^\top\mathbf{v} \leq y_i\mathbf{z}_i^\top\mathbf{v}' + \max_{i\in[n]}\|\mathbf{z}_{0:k}^{(i)}\|\epsilon \leq y_i\mathbf{z}_i^\top\mathbf{v}' + C\sqrt{k\ln(n/\delta)}\epsilon \leq y_i\mathbf{z}_i^\top\mathbf{v}' - 0.1,$$

where we set $\epsilon = 0.1/(C\sqrt{k\ln(n/\delta)})$. Therefore we must have

$$\mathbb{P}\left(\text{for every } \mathbf{v} \in \mathcal{S}, \ \sum_{i=1}^n \mathbb{1}\left[y_i\mathbf{z}_i^\top\mathbf{v} \leq -0.4\right] \geq \frac{n/8}{1 + \exp(\|\mathbf{v}^*\|)}\right)$$

$$\geq 1 - \exp\left(-\frac{n/32}{1 + \exp(\|\mathbf{v}^*\|)} + Ck\ln(1/\epsilon)\right) - 0.5\delta \geq 1 - \delta,$$

where in the last inequality we use the assumption that

$$n \geq C\big(1 + \exp(\|\mathbf{v}^*\|)\big)k\ln(k/\delta)$$

for a sufficiently large constant $C > 1$. We have completed the proof. $\qquad\square$

**Lemma C.4.** *Assume that $\mathbf{v}^*$ is $k$-sparse and*

$$n \geq C\big(1 + \exp(\|\mathbf{v}^*\|)\big)k\ln(k/\delta), \quad d \geq Cn\ln(n)\ln(1/\delta),$$

*where $C > 1$ is a large constant. Then with probability at least $1 - \delta$, we have*

$$\text{for every unit } \mathbf{v} \text{ such that } \max_{i\in[n]} y_i\mathbf{z}_i^\top\mathbf{v} < 0, \quad 1 - \frac{\langle\mathbf{v},\mathbf{v}^*\rangle}{\|\mathbf{v}^*\|} \geq \sqrt{\frac{1}{C\big(1 + \exp(\|\mathbf{v}^*\|)\big)}} \cdot \sqrt{\frac{n}{d}}.$$

*Proof of Lemma C.4.* Let $\mathbf{v}$ be an arbitrarily unit vector such that $\max_{i\in[n]} y_i\mathbf{z}_i^\top\mathbf{v} < 0$. Since $\mathbf{v}^*$ is $k$-sparse, without loss of generality, assume $\mathbf{v}_{k:\infty}^* = 0$, then

$$1 - \frac{\langle\mathbf{v},\mathbf{v}^*\rangle}{\|\mathbf{v}^*\|} = 1 - \frac{\langle\mathbf{v}_{0:k},\mathbf{v}_{0:k}^*\rangle}{\|\mathbf{v}_{0:k}^*\|} \geq 1 - \|\mathbf{v}_{0:k}\|.$$

It suffices to establish an upper bound on $\mathbf{v}_{0:k}^*$.

Define a set of indexes of the data that is significantly incorrectly classified by $\mathbf{v}_{0:k}$,

$$\mathcal{I} := \{i \in [n] : y_i\mathbf{z}_i^\top\mathbf{v}_{0:k} \leq -0.4\|\mathbf{v}_{0:k}\|\}$$

Then for each $i \in \mathcal{I}$, we must have

$$0 < y_i\mathbf{z}_i^\top\mathbf{v} = y_i\mathbf{z}_{0:k}^{(i)}\mathbf{v}_{0:k} + y_i\mathbf{z}_{k:\infty}^{(i)}\mathbf{v}_{k:\infty} \leq -0.4\|\mathbf{v}_{0:k}\| + y_i\mathbf{z}_{k:\infty}^{(i)}\mathbf{v}_{k:\infty}.$$

By Lemma C.3, we have

$$\Pr\left(|\mathcal{I}| \geq \frac{n/8}{1 + \exp(\|\mathbf{v}^*\|)}\right) \geq 1 - \delta.$$

According to (Hsu et al., 2021), we have

$$\Pr\left(\text{all } \big(y_i, \mathbf{z}_{k:\infty}^{(i)}\big)_{i\in\mathcal{I}} \text{ is support vector}\right) \geq 1 - \delta$$

assuming that $d \geq C|\mathcal{I}|\ln(\mathcal{I})\ln(1/\delta)$. Under this event, the max-margin direction is given by $\hat{\mathbf{v}} := \mathbf{Z}_{k:\infty}(\mathbf{Z}_{k:\infty}\mathbf{Z}_{k:\infty}^\top)^{-1}\mathbf{y}$, where $\mathbf{Z}_{k:\infty} = (\mathbf{z}_{k:\infty}^{(i)})_{i\in\mathcal{I}}$. It is clear that for every $i \in \mathcal{I}$, we have $\mathbf{z}_{k:\infty}^{(i)}\hat{\mathbf{v}} = y_i$. So we have

$$\max_{\|\mathbf{u}\|=1} \min_{i\in\mathcal{I}} y_i \mathbf{z}_{k:\infty}^{(i)}\mathbf{u} = \frac{1}{\|\hat{\mathbf{v}}\|} = \frac{1}{\sqrt{\mathbf{y}^\top(\mathbf{Z}_{k:\infty}\mathbf{Z}_{k:\infty}^\top)^{-1}\mathbf{y}}} \leq \sqrt{\frac{\|\mathbf{Z}_{k:\infty}\mathbf{Z}_{k:\infty}^\top\|_2}{\|\mathbf{y}\|^2}} = \sqrt{\frac{\|\mathbf{Z}_{k:\infty}\mathbf{Z}_{k:\infty}^\top\|_2}{n}}.$$

By Hoeffding's inequality, we have

$$\Pr\left(\mathbf{Z}_{k:\infty}\mathbf{Z}_{k:\infty}^\top \preceq 2d\mathbf{I}_n\right) \geq 1 - \exp(-C(d-n)) \geq 1 - \delta.$$

Under this event, we have $\max_{\|\mathbf{u}\|=1} \min_{i\in\mathcal{I}} y_i \mathbf{z}_{k:\infty}^{(i)}\mathbf{u} \leq \sqrt{2d/|\mathcal{I}|}$. Then we get

$$0.4\|\mathbf{v}_{0:k}\| \leq \min_{i\in\mathcal{I}} y_i \mathbf{z}_{k:\infty}^{(i)}\mathbf{v}_{k:\infty} \leq \|\mathbf{v}_{k:\infty}\|\sqrt{2d/|\mathcal{I}|}.$$

Since $\|\mathbf{v}\| = 1$, we must have

$$\|\mathbf{v}_{0:k}\| \leq \frac{1}{\sqrt{1 + 0.4|\mathcal{I}|/(2d)}}$$

Under the joint of the three events, which happens with probability at least $1 - 3\delta$, we have

$$\begin{aligned}
1 - \frac{\langle\mathbf{v},\mathbf{v}^*\rangle}{\|\mathbf{v}^*\|} &= 1 - \frac{\langle\mathbf{v}_{0:k},\mathbf{v}_{0:k}^*\rangle}{\|\mathbf{v}_{0:k}^*\|} \\
&\geq 1 - \|\mathbf{v}_{0:k}^*\| \\
&\geq 1 - \frac{1}{\sqrt{1 + 0.4|\mathcal{I}|/(2d)}} \\
&\geq C\sqrt{\frac{|\mathcal{I}|}{d}} \\
&\geq C\sqrt{\frac{1}{1 + \exp(\|\mathbf{v}^*\|)} \cdot \frac{n}{d}}.
\end{aligned}$$

Here $C > 1$ is a large constant and may vary line by line. We complete the proof by rescaling $\delta$. $\square$

*Proof of Theorem 4.2.* Define

$$\mathbf{v} = \boldsymbol{\Sigma}^{1/2}\mathbf{w}, \quad \mathbf{v}^* = \boldsymbol{\Sigma}^{1/2}\mathbf{w}^*, \quad \mathbf{z} = \boldsymbol{\Sigma}^{-1/2}\mathbf{x}, \quad d := \mathrm{rank}(\boldsymbol{\Sigma}).$$

Then $\mathbf{x}^\top\mathbf{w} = \mathbf{z}^\top\mathbf{v}$ and $\mathbf{x}^\top\mathbf{w}^* = \mathbf{z}^\top\mathbf{v}^*$. Without loss of generality, let $\|\mathbf{v}\| = 1$. Let $\theta$ be the angle between $\mathbf{v}$ and $\mathbf{v}^*$. We can apply Lemma C.4 to get

$$1 - \cos\theta = 1 - \frac{\langle\mathbf{v},\mathbf{v}^*\rangle}{\|\mathbf{v}^*\|} \gtrsim \sqrt{\frac{n}{d}} \gtrsim \frac{1}{\sqrt{\ln(n)\ln(1/\delta)}}.$$

We complete the proof by calling Lemma C.1. $\square$

# D. Early Stopping and $\ell_2$-Regularization

## D.1. Proof of Theorem 5.1

*Proof of Theorem 5.1.* We apply the $\ell_2$-regularized ERM $\mathbf{u}_\lambda$ with $\lambda = 1/(\eta t)$ as a comparator in Lemma 3.3. Recall that by the first-order stationary point condition, we have

$$-\nabla\mathcal{L}(\mathbf{u}_\lambda) = \lambda\mathbf{u}_\lambda = \frac{1}{\eta t}\mathbf{u}_\lambda.$$

Then we have

$$\frac{1}{2}\|\mathbf{w}_t - \mathbf{u}_t\|^2 - \frac{1}{2}\|\mathbf{u}_t\|^2 \leq \eta t\big(\widehat{\mathcal{L}}(\mathbf{u}_\lambda) - \widehat{\mathcal{L}}(\mathbf{w}_t)\big)$$
$$\leq \eta t\langle \nabla \mathcal{L}(\mathbf{u}_t), \mathbf{u}_t - \mathbf{w}_t\rangle$$
$$= -\langle \mathbf{u}_t, \mathbf{u}_t - \mathbf{w}_t\rangle,$$

where the two inequalities are by Lemma 3.3 and convexity, respectively. The above is equivalent to

$$\frac{1}{2}\|\mathbf{w}_t - \mathbf{u}_\lambda\|^2 \leq \langle \mathbf{u}_\lambda, \mathbf{w}_t\rangle - \frac{1}{2}\|\mathbf{u}_\lambda\|^2 \quad \Leftrightarrow \quad \|\mathbf{w}_t - \mathbf{u}_\lambda\|^2 \leq \frac{1}{2}\|\mathbf{w}_t\|^2. \tag{3}$$

To get the angle bound, we reformulate (3) as

$$2\langle \mathbf{u}_t, \mathbf{w}_t\rangle \geq \frac{1}{2}\|\mathbf{w}_t\|^2 + \|\mathbf{u}_t\|^2.$$

Then we get

$$\cos(\mathbf{w}_t, \mathbf{u}_t) := \frac{\langle \mathbf{u}_t, \mathbf{w}_t\rangle}{\|\mathbf{u}_t\| \cdot \|\mathbf{w}_t\|} \geq \frac{\frac{1}{2}\left(\frac{1}{2}\|\mathbf{w}_t\|^2 + \|\mathbf{u}_t\|^2\right)}{\|\mathbf{u}_t\| \cdot \|\mathbf{w}_t\|} \geq \frac{1}{\sqrt{2}},$$

where we use $(a+b)/2 \geq \sqrt{ab}$ for $a, b \geq 0$ in the last inequality. To get the norm bounds, we use triangle inequalities with (3) to get

$$\frac{1}{\sqrt{2}}\|\mathbf{w}_t\| \geq \|\mathbf{w}_t - \mathbf{u}_\lambda\| \geq \begin{cases} \|\mathbf{w}_t\| - \|\mathbf{u}_\lambda\|, \\ \|\mathbf{u}_\lambda\| - \|\mathbf{w}_t\|, \end{cases}$$

which implies

$$\frac{\sqrt{2}}{\sqrt{2}+1}\|\mathbf{u}_\lambda\| \leq \|\mathbf{w}_t\| \leq \frac{\sqrt{2}}{\sqrt{2}-1}\|\mathbf{u}_\lambda\|.$$

We complete the proof. $\qquad\qquad\qquad\qquad\qquad\qquad\qquad\qquad\qquad\qquad\qquad\qquad\qquad\qquad\qquad\qquad\square$

## D.2. Proof of Theorem 5.2

Recall that the dataset $(\mathbf{x}_i, y_i)_{i=1}^n$ is linearly separable. Define the margin and the maximum $\ell_2$-margin direction as

$$\gamma := \max_{\|\mathbf{w}\|=1} \min_{i \in [n]} y_i \mathbf{x}_i^\top \mathbf{w} > 0, \quad \tilde{\mathbf{w}} := \arg\max_{\|\mathbf{w}\|=1} \min_{i \in [n]} y_i \mathbf{x}_i^\top \mathbf{w}.$$

Both GD and $\ell_2$-regularization paths are rational invariant. Without loss of generality, we can assume $\tilde{\mathbf{w}} = \mathbf{e}_1$, where $(\mathbf{e}_i)_{i=1}^d$ denotes the canonical basis (Wu et al., 2023). That is to say, we will use the first coordinate of a vector to refer to its projection along the $\tilde{\mathbf{w}}$ direction. Under this convention, we use $(w_t, \bar{\mathbf{w}}_t)_{t \geq 0}$ and $(u_\nu, \bar{\mathbf{u}}_\nu)_{\nu \geq 0}$ to denote the optimization and regularization paths, respectively. We denote the data by

$$\mathbf{x}_i = (x_i, \bar{\mathbf{x}}_i), \quad y_i x_i \geq \gamma, \quad \bar{\mathbf{x}}_i \in \mathbb{R}^{d-1}.$$

Let the set of support vectors be

$$\mathcal{S} := \{i \in [n] : y_i \mathbf{x}_i^\top \tilde{\mathbf{w}} = \gamma\}.$$

By (Wu et al., 2023), the dataset $(\bar{\mathbf{x}}_i, y_i)_{i \in \mathcal{S}}$ is strictly non-separable under Assumption 2. In particular, by Definition 2 in (Wu et al., 2023), there exists $b > 0$ such that

$$\text{for every } \bar{\mathbf{w}} \in \mathbb{R}^{d-1}, \text{ there exists } i \in \mathcal{S} \text{ such that } y_i \bar{\mathbf{x}}_i^\top \bar{\mathbf{w}} \leq -b\|\bar{\mathbf{w}}\|. \tag{4}$$

Define

$$G(\bar{\mathbf{w}}) := \sum_{i \in \mathcal{S}} \exp(-y_i \bar{\mathbf{x}}_i^\top \bar{\mathbf{w}}).$$

Then $G(\cdot)$ is convex and $G(\bar{\mathbf{w}}) \geq 1$. By (4), the level set of $G(\bar{\mathbf{w}})$ is compact and $\bar{\mathbf{w}}^* := \arg\min G(\bar{\mathbf{w}})$ is finite.

The following lemma for the limiting GD path is from (Wu et al., 2023).

**Lemma D.1.** *Let* $\mathbf{w}_t := (w_t, \bar{\mathbf{w}}_t)$ *for* $t \geq 0$ *be the GD path with any fixed stepsize* $\eta > 0$. *Then under Assumption 2, we have* $w_t$ *is increasing and*

$$\lim_{t\to\infty} w_t = \infty, \quad \lim_{t\to\infty} \bar{\mathbf{w}}_t = \bar{\mathbf{w}}^*.$$

The following lemma characterizes the limiting $\ell_2$-regularization path.

**Lemma D.2.** *Let* $\mathbf{u}_\lambda = (u_\lambda, \bar{\mathbf{u}}_\lambda)$ *for* $\lambda > 0$ *be the* $\ell_2$-*regularization path. Then under Assumption 2, we have*

$$\lim_{\lambda\to 0} u_\lambda = \infty, \quad \lim_{\lambda\to 0} \bar{\mathbf{u}}_\lambda = \bar{\mathbf{w}}^*.$$

*Proof of Lemma D.2.* By the first order condition, we have

$$\lambda u_\lambda = -\frac{\mathrm{d}}{\mathrm{d}u}L(\mathbf{u}_\lambda) = \frac{1}{n}\sum_{i=1}^n \frac{y_i x_i}{1 + \exp(y_i x_i u_\lambda + y_i \bar{\mathbf{x}}_i^\top \bar{\mathbf{u}}_\lambda)},$$

$$\lambda \bar{\mathbf{u}}_\lambda = -\frac{\mathrm{d}}{\mathrm{d}\bar{\mathbf{u}}}L(\mathbf{u}_\lambda) = \frac{1}{n}\sum_{i=1}^n \frac{y_i \bar{\mathbf{x}}_i}{1 + \exp(y_i x_i u_\lambda + y_i \bar{\mathbf{x}}_i^\top \bar{\mathbf{u}}_\lambda)}.$$

We first show that $u_\lambda \to \infty$. Recall that $y_i x_i \geq \gamma$. Then we have

$$\begin{aligned}
\lambda u_\lambda &= \frac{1}{n}\sum_{i=1}^n \frac{y_i x_i}{1 + \exp(y_i x_i u_\lambda + y_i \bar{\mathbf{x}}_i^\top \bar{\mathbf{u}}_\lambda)} \\
&\geq \frac{\gamma}{n}\sum_{i=1}^n \frac{1}{1 + \exp(y_i x_i u_\lambda + y_i \bar{\mathbf{x}}_i^\top \bar{\mathbf{u}}_\lambda)} \\
&\geq \frac{\gamma}{2n}\sum_{i=1}^n \exp(-y_i x_i u_\lambda - y_i \bar{\mathbf{x}}_i^\top \bar{\mathbf{u}}_\lambda) \\
&\geq \frac{\gamma}{2n}\sum_{i\in\mathcal{S}} \exp(-y_i x_i u_\lambda - y_i \bar{\mathbf{x}}_i^\top \bar{\mathbf{u}}_\lambda) \\
&= \frac{\gamma}{2n} \exp(-\gamma u_\lambda)G(\bar{\mathbf{u}}_\lambda) \\
&\geq \frac{\gamma}{2n} \exp(-\gamma u_\lambda),
\end{aligned}$$

where the last inequality is because $G(\cdot) \geq 1$. The above bound implies that $u_\lambda \to \infty$ as $\lambda \to 0$.

We next show that $\bar{\mathbf{u}}_\lambda$ is bounded. Define

$$\mathcal{N} := \{i \in [n] : y_i \bar{\mathbf{x}}_i^\top \bar{\mathbf{u}}_\lambda < 0\}, \quad \mathcal{P} := \{i \in [n] : y_i \bar{\mathbf{x}}_i^\top \bar{\mathbf{u}}_\lambda \geq 0\}.$$

Then $\mathcal{N}$ is nonempty by (4). Moreover, there exists $i^* \in \mathcal{N} \cap \mathcal{S}$ such that $y_{i^*} \bar{\mathbf{x}}_{i^*}^\top \bar{\mathbf{u}}_\lambda \leq -b\|\bar{\mathbf{u}}\|$ by (4). Note that $y_{i^*} x_{i^*} = \gamma$ by the definition of $\mathcal{S}$. Then by definition, we have

$$\begin{aligned}
\lambda\|\bar{\mathbf{u}}\|^2 &= \frac{1}{n}\sum_{i=1}^n \frac{y_i \bar{\mathbf{x}}_i^\top \bar{\mathbf{u}}}{1 + \exp(y_i x_i u_\lambda + y_i \bar{\mathbf{x}}_i^\top \bar{\mathbf{u}}_\lambda)} \\
&= \frac{1}{n}\sum_{i\in\mathcal{N}} \frac{y_i \bar{\mathbf{x}}_i^\top \bar{\mathbf{u}}}{1 + \exp(y_i x_i u_\lambda + y_i \bar{\mathbf{x}}_i^\top \bar{\mathbf{u}}_\lambda)} + \frac{1}{n}\sum_{i\in\mathcal{P}} \frac{y_i \bar{\mathbf{x}}_i^\top \bar{\mathbf{u}}}{1 + \exp(y_i x_i u_\lambda + y_i \bar{\mathbf{x}}_i^\top \bar{\mathbf{u}}_\lambda)} \\
&\leq \frac{1}{n}\sum_{i\in\mathcal{N}} \frac{y_i \bar{\mathbf{x}}_i^\top \bar{\mathbf{u}}}{1 + \exp(y_i x_i u_\lambda)} + \frac{1}{n}\sum_{i\in\mathcal{P}} y_i \bar{\mathbf{x}}_i^\top \bar{\mathbf{u}} \exp(-y_i x_i u_\lambda - y_i \bar{\mathbf{x}}_i^\top \bar{\mathbf{u}}_\lambda) \\
&\leq \frac{1}{n} \frac{y_{i^*} \bar{\mathbf{x}}_{i^*}^\top \bar{\mathbf{u}}}{1 + \exp(y_{i^*} x_{i^*} u_\lambda)} + \frac{1}{n}\sum_{i\in\mathcal{P}} \exp(-y_i x_i u_\lambda)
\end{aligned}$$

$$\leq \frac{1}{n} \frac{-b\|\bar{\mathbf{u}}\|}{1 + \exp(\gamma u_\lambda)} + \exp(-\gamma u_\lambda),$$

where the second inequality is by $e^t t \leq 1$. The above inequality implies

$$b\|\bar{\mathbf{u}}_\lambda\| \leq n\big(1 + \exp(\gamma u_\lambda)\big) \exp(-\gamma u_\lambda) \leq 2n,$$

where the last inequality is because $u_\lambda > 0$. This shows that $\bar{\mathbf{u}}_\lambda$ is bounded.

Now we prove an upper bound on $u_\lambda$. Since $\bar{\mathbf{u}}_\lambda$ is bounded, we know $\sum_{i=1}^n \exp(-y_i \bar{\mathbf{x}}_i^\top \bar{\mathbf{u}}_\lambda)$ is bounded by a constant $F_{\max}$. Let $X := \max_i \|\mathbf{x}_i\|$. Then we have

$$\lambda u_\lambda = \frac{1}{n} \sum_{i=1}^n \frac{y_i x_i}{1 + \exp(y_i x_i u_\lambda + y_i \bar{\mathbf{x}}_i^\top \bar{\mathbf{u}}_\lambda)}$$

$$\leq \frac{X}{n} \sum_{i=1}^n \frac{1}{1 + \exp(y_i x_i u_\lambda + y_i \bar{\mathbf{x}}_i^\top \bar{\mathbf{u}}_\lambda)}$$

$$\leq \frac{X}{n} \sum_{i=1}^n \exp(-y_i x_i u_\lambda - y_i \bar{\mathbf{x}}_i^\top \bar{\mathbf{u}}_\lambda)$$

$$\leq \frac{X}{n} \sum_{i=1}^n \exp(-\gamma u_\lambda - y_i \bar{\mathbf{x}}_i^\top \bar{\mathbf{u}}_\lambda)$$

$$\leq \frac{X F_{\max}}{n} \exp(-\gamma u_\lambda),$$

which implies that

$$\lambda \exp(\gamma u_\lambda) \leq \frac{X F_{\max}}{n u_\lambda} \to 0. \tag{5}$$

Finally, we show $\bar{\mathbf{u}}_\lambda \to \bar{\mathbf{w}}^*$. Recall the definition of $\bar{\mathbf{u}}_\lambda$. Let us take $\lambda \to 0$ and use (5), the boundedness of $\bar{\mathbf{u}}_\lambda$, and that

$$y_i x_i > \gamma \text{ for } i \notin \mathcal{S}, \quad y_i x_i = \gamma \text{ for } i \in \mathcal{S},$$

then we get

$$0 = \lim_{\lambda \to 0} \lambda \bar{\mathbf{u}}_\lambda \exp(\gamma u_\lambda)$$

$$= \lim_{\lambda \to 0} \frac{1}{n} \sum_{i=1}^n \frac{y_i \bar{\mathbf{x}}_i \exp(\gamma u_\lambda)}{1 + \exp(y_i x_i u_\lambda + y_i \bar{\mathbf{x}}_i^\top \bar{\mathbf{u}}_\lambda)}$$

$$= \lim_{\lambda \to 0} \frac{1}{n} \sum_{i \in \mathcal{S}} \frac{y_i \bar{\mathbf{x}}_i \exp(\gamma u_\lambda)}{1 + \exp(\gamma u_\lambda + y_i \bar{\mathbf{x}}_i^\top \bar{\mathbf{u}}_\lambda)}$$

$$= \lim_{\lambda \to 0} \frac{1}{n} \sum_{i \in \mathcal{S}} \frac{y_i \bar{\mathbf{x}}_i \exp(\gamma u_\lambda)}{\exp(\gamma u_\lambda + y_i \bar{\mathbf{x}}_i^\top \bar{\mathbf{u}}_\lambda)}$$

$$= \lim_{\lambda \to 0} \frac{1}{n} \sum_{i \in \mathcal{S}} \frac{y_i \bar{\mathbf{x}}_i}{\exp(y_i \bar{\mathbf{x}}_i^\top \bar{\mathbf{u}}_\lambda)}.$$

Thus $\bar{\mathbf{u}}_0$ satisfies the first-order stationary condition of $G(\cdot)$. So we must have $\bar{\mathbf{u}}_0 = \bar{\mathbf{w}}^*$. This completes our proof. $\qquad \square$

*Proof of Theorem 5.2.* The proof is a direct consequence of the above lemmas.

Note that $u_\lambda$ is continous, $u_\infty = 0$, and $\lim_{\lambda \to 0} u_\lambda = \infty$. Moreover $w_0 = 0$, $w_t$ is increasing, and $w_t \to \infty$. Thus for each $t$ we can choose $\lambda(t)$ such that $u_\lambda = w_t$ and $\lambda(t) \to 0$. Then we have

$$\|\mathbf{u}_{\lambda(t)} - \mathbf{w}_t\| = \|\bar{\mathbf{u}}_{\lambda(t)} - \bar{\mathbf{w}}_t\| \leq \|\bar{\mathbf{u}}_{\lambda(t)} - \bar{\mathbf{w}}^*\| + \|\bar{\mathbf{w}}_t - \bar{\mathbf{w}}^*\| \to 0.$$

This completes the proof. $\qquad \square$

## D.3. Proof of Theorem 5.3

*Proof of Theorem 5.3.* As the dataset is linearly separable and we only care about the asymptotic, without loss of generality, we can consider the exponential loss instead of the logistic loss. In what follows, we focus on analyzing gradient flow. Our argument applies to gradient descent with any fixed stepsize $\eta > 0$.

Denote $\mathbf{w} = (w^{(1)}, w^{(2)})$. Then the empirical risk can be written as

$$\widehat{\mathcal{L}}(w^{(1)}, w^{(2)}) := \frac{1}{2}\big(\exp(-\gamma w^{(1)}) + \exp(-\gamma w^{(1)} - \gamma_2 w^{(2)})\big).$$

Note that the GF and $\ell_2$-regularization paths under the $\widehat{\mathcal{L}}$ are the same as those under $\ln \widehat{\mathcal{L}}$ up to a rescaling of the time and regularization strength. It suffices to compared the GF and $\ell_2$-regularization paths under $\ln \widehat{\mathcal{L}}$, where

$$\ln \widehat{\mathcal{L}}(w^{(1)}, w^{(2)}) = -\gamma w^{(1)} + \ln\big(1 + \exp(-\gamma_2 w^{(2)})\big) - \ln(2).$$

The GF path is given by

$$\mathrm{d}w_t^{(1)} = \gamma \mathrm{d}t, \quad \mathrm{d}w_t^{(2)} = \gamma_2 \frac{\exp(-\gamma_2 w^{(2)})}{1 + \exp(-\gamma_2 w^{(2)})}\mathrm{d}t, \quad w_0^{(1)} = w_0^{(2)} = 0.$$

Solving the ODEs, we get

$$w_t^{(1)} = \gamma t, \quad \left| w_t^{(2)} - \frac{\ln(1 + \gamma_2^2 t)}{\gamma_2} \right| \leq C, \quad t \geq 0,$$

for some constant $C > 0$. The $\ell_2$-regularization path is given by

$$-\gamma + \lambda u_\lambda^{(1)} = 0, \quad -\gamma_2 \frac{\exp(-\gamma_2 u^{(2)})}{1 + \exp(-\gamma_2 u^{(2)})} + \lambda u_\lambda^{(2)} = 0.$$

Then we get

$$u_\lambda^{(1)} = \frac{\gamma}{\lambda}, \quad \lambda \geq 0$$

$$\left| u_\lambda^{(2)} - \frac{\ln(\gamma_2^2/\lambda) - \ln\ln(\gamma^2/\lambda)}{\gamma_2} \right| \leq C, \quad \lambda \leq \gamma_2^2/e,$$

As $t \to \infty$ and $\lambda \to 0$, the two paths tend to infinity in both directions. However, due to the rate mismatch, it is impossible to match $\mathbf{w}_t$ with $\mathbf{u}_\lambda$ in both directions at the same time. So their $\ell_2$-distance has to be infinite asymptotically. $\square$

