# OpenReview forum: "Benefits of Early Stopping in Gradient Descent for Overparameterized Logistic Regression"
_ICML.cc/2025/Conference — ICML 2025 poster_

### Official Review · Reviewer_Ryys · 2025-03-11

**Overall Recommendation:** 4

**Summary:**

This paper investigate the importance of early stopping in well-specified high-dimensional logistic regression. They demonstrate that the early stopping ensures generalization in terms of excess logistic risk, while interpolator diverges. These results emphasize the need of early stopping in overparameterized models.

**Claims And Evidence:**

Yes.

**Essential References Not Discussed:**

NA.

**Experimental Designs Or Analyses:**

NA.

**Methods And Evaluation Criteria:**

Yes.

**Other Comments Or Suggestions:**

NA

**Other Strengths And Weaknesses:**

###  Strengths

The paper is well-written and easy to follow.
The techinical results are comprehensive that both upper bounds for early-stopped GD and lower bounds for interpolators are given, which are persuasive.
Moreover, the connection between early stopping and explicit $l^2$ regularization is discussed.

### Weakness

The early-stopping time is oracle-based and lacks explicit bounds, which will limit the practical utility of the results.

**Questions For Authors:**

1. What are the insights of selecting $k$ in the main theorems? How the "variance" and the "bias" are connected to $k$?

2. Can the results in this paper be extended to generalized linear models or M-estimators?

**Relation To Broader Scientific Literature:**

This paper considers early-stopped gradient descent in logistic regression, while most of the previous works focus on linear regression.
The results in this paper extend the previous results to the setting of logistic regression, which is novel.
These results contribute to understandings of early stopping in over-parameterized statistical problems.

**Theoretical Claims:**

The proofs seem to be solid, but I have not checked all the details.

---

> ### Author Rebuttal · Authors · 2025-03-31
>
> Thank you for supporting our paper. We answer your questions as follows.
>
> ---
> **Q1.** “The early-stopping time is oracle-based and lacks explicit bounds, which will limit the practical utility of the results.”
>
> **A1.** Our aim is to understand the benefits of early stopping. We believe designing a practical criterion for early stopping, despite being an important question, is beyond the scope of this paper.
>
>
> ---
> **Q2.** “What are the insights of selecting $k$ in the main theorems? How the "variance" and the "bias" are connected to $k$?”
>
> **A2.** Intuitively, $k$ determines the number of dimensions in which early stopped GD is able to learn the true parameter $w\^\*$. Moreover, early stopped GD ignores the remaining dimensions and pays an “approximation” error. We choose $k$ to minimize the upper bounds.
>
> In Theorem 3.1, the stopping time is relatively small, at which point GD first hits an empirical risk of $\\hat L(w\_{0:k}\^\*)$. Thus, the total parameter movement is relatively small, and the bias error tends to be large. In Theorem 3.2, the stopping time is relatively large, at which point GD hits an empirical risk of $\\hat L(w\^*)$. Thus, the total parameter movement is large, and the variance error tends to be large.
>
> We will clarify these in the revision.
>
> ---
> **Q3.** “Can the results in this paper be extended to generalized linear models or M-estimators?”
>
> **A3.** We believe that part of our results can be extended beyond logistic regression. For example, the proof of Theorem 3.1 can be adapted to other loss functions that are convex, smooth, and Lipschitz. We will comment on this as a future direction.

---

### Official Review · Reviewer_KGpn · 2025-03-11

**Overall Recommendation:** 3

**Summary:**

The paper studies the impact of early stopping in gradient descent (GD) for overparameterized logistic regression. It demonstrates that, in this setting, early-stopped GD is well-calibrated and achieves lower risk and zero-one risk compared to GD at convergence. Additionally, the paper establishes a connection between the implicit regularization of GD and explicit regularization.

# Update after rebuttal

The authors have addressed my concerns. I keep my positive score

**Claims And Evidence:**

Yes, the authors prove all of their results.

**Essential References Not Discussed:**

The paper discusses all of the essential references.

**Experimental Designs Or Analyses:**

N.A

**Methods And Evaluation Criteria:**

N.A

**Other Comments Or Suggestions:**

N.A

**Other Strengths And Weaknesses:**

Strengths:

1. The paper is well-written and clearly structured.

2. It thoroughly discusses related work and provides detailed comparisons with its own results.

3. The findings on well-calibration and improved generalization with respect to 0-1 risk, compared to interpolating GD, are both interesting and somewhat surprising.

4. The paper extensively discusses the limitations of its results.

Weaknesses:

1.The technique used to derive the logistic risk upper bounds relies heavily on the prior work of Telgarsky (2022).

2.The results are based on an optimal stopping time. In particular, the authors acknowledge that "early-stopped GD in Theorem 3.1 is not a practical algorithm". It would be beneficial to present results that explicitly depend on the stopping time or provide an example of a stopping time for which the bounds hold.

3. In cases where the results are comparable to prior work, the bounds obtained in this paper are sometimes weaker.

**Questions For Authors:**

Are there scenarios where the optimal stopping time is impractically large, such as exponential in $ d $ or $ n $? If so, what happens to the risk of GD when it is stopped after $\text{poly}(d,n)$ iterations? Does it still benefit from early stopping in such cases?

**Relation To Broader Scientific Literature:**

Most of the paper's main contributions are not directly comparable to previous work. The most relevant studies primarily focus on asymptotic GD, the setting of separable data, or scenarios where the loss function is the squared loss.

**Theoretical Claims:**

No. I focused on reading the main text.

---

> ### Author Rebuttal · Authors · 2025-03-31
>
> We thank the reviewer for their comments. We address your questions below.
>
> ---
> **Q1.** “The technique used to derive the logistic risk upper bounds relies heavily on the prior work of Telgarsky (2022).”
>
> **A1.** While our Lemma 3.3 appears in [Telgarsky 2022] (and other prior works as mentioned), it seems unfair to call it a “weakness”, as new research builds upon existing results. We would also like to point out that our logistic risk upper bounds require careful statistical control around the population optimum since we work in an overparameterized regime, which is beyond [Telgarsky 2022]. Besides, we show how early stopped GD separates from asymptotic GD and how it explicitly connects to $\ell_2$-regularization. None of these results appears in [Telgarsky 2022].
>
> ---
> **Q2.** “...It would be beneficial to present results that explicitly depend on the stopping time or provide an example of a stopping time for which the bounds hold.”
>
> **A2.** One can compute an upper bound on the stopping time using standard optimization and concentration tools. Specifically, $\\hat L$ is smooth and convex, so GD converges in $O(1/t)$ rate. Moreover, we can compute $\\hat L(w\^*\_{0:k})$ using concentration bounds. In this way we can compute a stopping time upper bound.
>
> Once we obtain the upper bound on the stopping time, we can state the theorem as “there exists a $t$ no larger than XX, such that…”. This gives examples of stopping time for our bounds to hold. We will comment on this in the revision. However, we prefer to maintain our original theorem statement as it is cleaner.
>
> ---
> **Q3.** “In cases where the results are comparable to prior work, the bounds obtained in this paper are sometimes weaker.”
>
> **A3.** As discussed in Lines 397-425, no prior results strictly dominates our results; the work that obtained sharper rates also needs stronger assumptions than ours. While there is still room to improve our upper bounds (as discussed in the end of Section 3.2), we believe our contribution is already very significant by establishing the benefits of early stopping in logistic regression.
>
> ---
> **Q4.** “Are there scenarios where the optimal stopping time is impractically large, such as exponential in $d$ or $n$? If so, what happens to the risk of GD when it is stopped after $\mathsf{poly}(d,n)$ iterations? Does it still benefit from early stopping in such cases?”
>
> **A4.** Note that our results allow $d=\infty$, but the optimal stopping time is always finite as hinted by our negative results for asymptotic GD. Therefore, the stopping time is generally small compared to $d=\infty$, and it is meaningless to discuss “exponential or polynomial in $d$”, which is infinite.
>
> Moreover, as discussed in **A2**, the stopping time is at most polynomial in $n$ by optimization and concentration arguments. Note that the hidden constant factors might be instance-dependent. Again, there is no exponential dependence.

---

### Official Review · Reviewer_MdqZ · 2025-03-13

**Overall Recommendation:** 4

**Summary:**

This paper studies the problem of early stopping in logistic regression. It considers two metrics: the zero classification accuracy and the logistic loss. The paper shows that the excess zero one loss is bounded by the excess logistic loss. Building on this the paper shows that there exists an early stopped model that has zero excess logistic loss (asymptotically). Hence the model is well calibrated and consistent as well. On the other hand the paper also shows the existence of a setup where the converged solution has infinite excess logistic loss and that the excess zero one loss is at least a constant.

**Claims And Evidence:**

The paper is a theory paper and presents a variety of theoretical statements that support their claims. However, I have some questions about some of the statements and their proofs.

Quoting from the paper, it says that "The empirical risk $\hat{\mathcal{L}}(w_t) decreases monotonically for $t \ge 0$. Thus there exists a stopping time $t$ such that $\hat{\mathcal{L}}(w_t) \le  \hat{\mathcal{L}} (w^∗_{0:k} \le \hat{\mathcal{L}}(w_{t-1})$."

However, being monotonically decreasing does not mean that $\hat{\mathcal{L}}(w_t)$ is eventually smaller than $\hat{\mathcal{L}} (w^∗_{0:k}$. I tried checking the proof, but I could not find the part of the proof where the assertion that there exists a $t$ such that $\hat{\mathcal{L}}(w_t) \le  \hat{\mathcal{L}} (w^∗_{0:k} $ is proven.

Theorem 3.2 then assume the existence of such a time $t$. But the description in the paper seems to assume that such a time $t$ must exist.


--------

My current score is primarily due to the exists of $t$ such that $\hat{\mathcal{L}}(w_t) \le  \hat{\mathcal{L}} (w^∗_{0:k} $. I believe that the exietnce of such a $t$ needs to be proven to complete the story. Otherwise the current story is if there is a good early stopping point it is good to stop because the cobverged solution can be bad versus early stopping is good.

**Essential References Not Discussed:**

I think the paper [A] is an important recent paper that is quite relevant to many of the ideas discussed in the paper (early stopping, implicit regularisation, connection to explicit regularization) is missing

[A] Sonthalia, Rishi, Jackie Lok, and Elizaveta Rebrova. "On regularization via early stopping for least squares regression." arXiv preprint arXiv:2406.04425 (2024).

**Experimental Designs Or Analyses:**

No Experiments.

**Methods And Evaluation Criteria:**

There are no experiments.

**Other Comments Or Suggestions:**

N/A

**Other Strengths And Weaknesses:**

**Strengths**

The prose in between theorem statements is very well written. In fact, it is amongst the best I have ever seen.

The duality of Theorem 3.1 and 4.1 is quite surprising. In particular, in light of Theorem 4.2. One would imagine that things break down in the limit due to the norm going to infinity. However, Theorem 4.2 says that is not the case.

The conenction to ridge regularized version is also interesting.

**Weakness**

I think the theorem statements could be sharper. Words like - thus should not appear in a theorem statement.

**Questions For Authors:**

All of the current analyses are for the case when we initialize at zero. What if we initialized somewhere else, or did it have a random initialization?

just to confirm the probabilities in Theorems 3.1 and 3.2 aree with respect to the sampling of the data?

**Relation To Broader Scientific Literature:**

The relationship to the broader scientific literature is well established, and understanding the simplicit bias of gradient descent and early stopping in different scenarios is an important problem.

Earlying stopping and GD have been mostly studied for linear regression. The case for logistic regression is less well understood. Hence the duality of Theorems 3.1 and 4.1 is quite interesting in the loogistic regression setting.

**Theoretical Claims:**

There are a variety of theoretical claims. I did not check any of the proofs except for reading the lemma statements in Section C.1 and then the proof of Theorem 3.1 given the lemmas.

However, I think there is an issue with the proof. Please see my comments in the claims and evidence section.

---

> ### Author Rebuttal · Authors · 2025-03-31
>
> Thank you for your comments. We address your concerns as follows.
>
> ---
> **Q1.** “My current score is primarily due to the exists of $t$ such that $\\hat L(w\_t) \\le \\hat L(w\_\{0:k\}\^\*)$. I believe that the existence of such a $t$ needs to be proven to complete the story.”
>
> **A1.** The existence of a $t$ such that $\\hat L(w\_t) \\le \\hat L(w\_\{0:k\}\^\*)$ is guaranteed because
> * GD converges to a minimizer of the empirical risk $\\hat L(\\cdot)$;
> * and $\\min \\hat L(\\cdot)  < \\hat L(w\_\{0:k\}\^\*)$.
>
> So there exists a $t$ such that $\\hat L(w\_t) \\le \\hat L(w\_\{0:k\}\^\*)$.
> The second bullet is clear. We explain the first bullet as follows.
>
> Note that $\\hat L(\\cdot)$ is smooth and convex. Classical optimization theory guarantees the global convergence of GD in this case. Specific to logistic regression, this is proved in, for example, Theorem 1.1 in [Ji & Telgarsky 2018], with a precise convergence rate. We also see this by applying Lemma 3.3 with suitable $u$ and large enough $t$.
>
> This explanation should address your concerns. But let us know if you have further questions regarding this. We will add these discussions in the revision.
>
> ---
> **Q2.** “I think the paper [A] is an important recent paper that is quite relevant to many of the ideas discussed in the paper (early stopping, implicit regularisation, connection to explicit regularization) is missing”
>
> **A2.** Thanks for pointing out the missing reference. We will cite and discuss it in the revision.
>
> ---
> **Q3.** “I think the theorem statements could be sharper. Words like - thus should not appear in a theorem statement.”
>
> **A3.** We will polish our theorem statements according to your suggestions. Thanks.
>
> ---
> **Q4.** “All of the current analyses are for the case when we initialize at zero. What if we initialized somewhere else, or did it have a random initialization?”
>
> **A4.** When the initialization $w\_0$ is nonzero, in the bounds, $w\^\*$ should be replaced by $w\^\* - w\_0$. This can be seen from the proof of the theorems. We will clarify this in the revision.
>
> ---
> **Q5.** “just to confirm the probabilities in Theorems 3.1 and 3.2 agree with respect to the sampling of the data?”
>
> **A5.** Yes.

---

> > ### Comment · Reviewer_MdqZ · 2025-04-01
> >
> > Thank you for the clarification. I have increased my score.

---

### Official Review · Reviewer_Vcgf · 2025-03-24

**Overall Recommendation:** 3

**Summary:**

This paper theoretically examines the additional regularization bias introduced by early stopping in logistic regression. The authors first demonstrate that for any well-specified logistic regression problem, gradient descent (GD) with oracle-based early stopping is well-calibrated and statistically consistent. They then establish lower bounds on test risk, calibration error, and zero-one error for asymptotic GD, showing that it is inconsistent and has worse sample complexity compared to early-stopped GD. Additionally, they derive bounds on the angular differences between the regularization path and the GD path for any convex and smooth objective. Finally, they analyze asymptotic bounds for logistic regression with linearly separable data, showing that under a sufficient condition on the data, both paths converge.

**Update after rebuttal**

I thank the authors for engaging during the rebuttal period. They addressed most of my key questions and agreed to make appropriate additions to the paper. Overall, I believe that it is a good paper, and improves our understanding of early-stopped gradient descent, and I maintain my positive rating.

**Claims And Evidence:**

The claim on calibration via early stopping is well supported by Theorems 3.1 and 3.2. The negative results on poor calibration and the inconsistency of asymptotic GD are also established in Theorems 4.1 and 4.2, along with the need for exponential sample complexity for achieving zero-one error. However, I have some concerns regarding the comparison between Theorems 3.1 (3.2) and Theorems 4.1 and 4.2—see the questions section. Finally, Theorems 5.1 and 5.2 establish the global and asymptotic connection between GD and the regularization path.

**Essential References Not Discussed:**

Looks good and thorough to me.

**Experimental Designs Or Analyses:**

N/A, Look at the response to Methods and Evaluation Criteria.

**Methods And Evaluation Criteria:**

It’s mainly a theory paper. I don’t think this question is very valid, but the toy experiment in Figure 1 matches the theoretical claims in the paper.

**Other Comments Or Suggestions:**

N/A

**Other Strengths And Weaknesses:**

The paper studies a very important (and understudied) problem from a theoretical point of view. It is well-written in general, easy to read and follow.

**Questions For Authors:**

1. Thm 3.1 and 3.2 provide early stopping risk bounds for any (including overparameterized) well-specified logistic regression problem. Lines 266-273, discuss application of these bounds to a trivial, under-parameterized case. How do these bounds perform in the overparameterized case? $\Sigma$ needs to be carefully chosen to control the tail of EVs and the trace. For a meaningful comparison with the negative results in Section 4 on asymptotic GD, it would be useful to determine when Thm 3.1 and Thm 3.2 provide non-vacuous bounds and how common these settings are.


2. For Thm 4.2, what role does the $k$-sparsity condition play? Is this the key condition needed to prove inconsistency? Please provide some intuition based on the proof. Additionally, linking it to the previous question, can you provide an example (or a class of examples based on data covariance) where Thm 3.1 and Thm 3.2 yield non-vacuous bounds while the negative results in Thm 4.2 still hold?



3. Regarding conjectures at the end of page 6 and page 8 (left column end)—what justifies these claims? For the first, do you have any empirical results supporting it? For the second, Thm 5.2 and Thm 5.3 merely represent convergence under a sufficient condition and a counter example, the conjecture seems strong—do you have additional reasoning to support it?

**Relation To Broader Scientific Literature:**

The paper builds on the well-established theory of implicit regularization in GD for logistic regression by theoretically demonstrating the additional benefits of early stopping. While the generalization benefits of early-stopped GD have been studied in much greater detail for linear regression (for example, vanishing excess risk with early stopped GD despite no benign overfitting with $\ell_2$ norm interpolator (see references on column 1, page 2)), they are less well understood for logistic regression. This work addresses an important gap in the literature by showing that early stopped GD can achieve good generalization (vanishing excess risk), despite a statistically inconsistent interpolator, similar to results in linear regression.

**Theoretical Claims:**

No, I did not read the proofs of the theorems.

---

> ### Author Rebuttal · Authors · 2025-03-31
>
> Thank you for your detailed comments. We address your questions below.
>
> ---
> **Q1.** “...Lines 266-273, discuss application of these bounds to a trivial, under-parameterized case. How do these bounds perform in the overparameterized case?... For a meaningful comparison with the negative results in Section 4 on asymptotic GD, it would be useful to determine when Thm 3.1 and Thm 3.2 provide non-vacuous bounds and how common these settings are.”
>
> **A1.** In Lines 205-214 of the right column and Lines 272-274, we discuss our bounds in an overparameterized setting satisfying the standard source and capacity conditions, where our bounds are non-vacuous.
>
> Our comparison holds in general and common settings. In what follows, we explain this using Theorems 3.1 and 4.1 as examples. For every well-specified logistic regression problem (see Assumption 1), Theorem 3.1 yields a vanishing excess risk for early stopped GD as discussed in Lines 190-203 right. For the same set of problems, Theorem 4.1 implies GD without early stopping is inconsistent for logistic loss and poorly calibrated, as discussed in Lines 279-283 of the right column. Note that this comparison is agnostic to dimension and holds in the overparameterized regime as well.
>
> ---
> **Q2.** “For Thm 4.2, what role does the $k$-sparsity condition play? Is this the key condition needed to prove inconsistency? Please provide some intuition based on the proof. … can you provide an example (or a class of examples based on data covariance) where Thm 3.1 and Thm 3.2 yield non-vacuous bounds while the negative results in Thm 4.2 still hold?”
>
> **A2.** The $k$-sparsity is a sufficient condition to establish the sample complexity lower bound in Theorem 4.2. Note that Theorem 4.2 provides a lower bound on the excess zero-one error, but this does not imply inconsistency. For the inconsistency results in Theorem 4.1, we do not need the $k$-sparsity condition.
>
> The intuition behind Theorem 4.2 is that there are $k$ informative dimensions and a lot more uninformative dimensions. Since $n\gg k$, the training set cannot be separated purely using the $k$ informative dimensions. Thus, interpolators must use the uninformative dimensions to separate the data, leading to the risk lower bound. This explains the role of the $k$-sparsity in Theorem 4.2.
>
> We discuss in Lines 313-319 for situations where Theorems 3.1 and 3.2 yield non-vacuous bounds while the negative results in Thm 4.2 still hold. Moreover, the example discussed in Lines 205-214 of the right column and Lines 272-274 also suffices. We will make this clear in the revision.
>
> ---
> **Q3.** “Regarding conjectures at the end of page 6 and page 8 (left column end)—what justifies these claims? For the first, do you have any empirical results supporting it? For the second, Thm 5.2 and Thm 5.3 merely represent convergence under a sufficient condition and a counter example, the conjecture seems strong—do you have additional reasoning to support it?”
>
> **A3.** We discuss evidence/intuitions for the two conjectures as follows.
>
> For the first one, note that in Figure 1(b), the test zero-one error keeps increasing even after reaching interpolation (training zero-one error becomes zero). This suggests that the zero-one error of the maximum $\\ell\_2$-margin interpolator (this is when $t\to\\infty$) should be higher than an oblivious interpolator. Our conjecture is partly motivated by this observation.
>
> The reasoning behind the second conjecture is as follows. Note that Assumption 2 implies that the dataset projected perpendicular to the max-margin directions (called “projected dataset”) is strictly nonseparable (see Lemma 3.1 in Wu et al., 2023). This is the only property used in Theorem 5.2. Moreover, in Theorem 5.3, the “projected dataset” is nonseparable but with margin zero – we conjecture this property is sufficient for Theorem 5.3 to hold. Now for a generic separable dataset, we check the “projected dataset”:
> - if it is strictly non-separable, Theorem 5.2 holds;
> - if it is non-separable but with margin zero, we conjecture Theorem 5.3 holds;
> - otherwise it is separable (with positive margin), we decompose the dataset recursively.
> This is the reasoning behind our conjecture.
>
> We will add these discussions in the revision.

---

> > ### Comment · Reviewer_Vcgf · 2025-04-05
> >
> > Thank you for your response to my questions. Most of my questions have been answered, please add clarifications in the paper wherever discussed above. I have a few more questions: on parsing again, it feels like the connection of Section 5 to the rest of the paper is weak. Why do I care about the distance between regularization and GD paths (sure a complete characterization is good, but why is it important to the message of the rest of the sections)? For error, the distance doesn't matter, only correlation does, which is always 0, asymptotically. However, for calibration, distance does matter. Can we see an example where the two paths diverge (like in Thm 5.3), and one of them achieves better calibration?

---

> > > ### Author Response · Authors · 2025-04-06
> > >
> > > Thanks for confirming most of your questions were resolved—we will incorporate clarifications into the paper. Below we respond to your follow-up questions.
> > >
> > > **Role of Section 5**. We believe our results in Section 5 are closely tied to the rest of the paper. Sections 3 and 4 demonstrate that early stopping carries a certain regularization effect that benefits its statistical performance. This regularization is, however, implicit. In Section 5, we attempt to provide some intuitions of the implicit regularization of early stopping by establishing its connections to an explicit $\ell_2$-regularization. We will revise the text to better motivate its relevance and clarify the connection.
> > >
> > > **Importance of studying paths distance**. If GD and regularization paths were uniformly and absolutely close, one could argue that early stopping fully mimics $\ell_2$-regularization. However, our results show that while the two paths are relatively close in general, absolute closeness only holds in special cases. This suggests that the implicit regularization induced by early stopping might not be entirely equivalent to $\ell_2$-regularization (despite being highly similar and comparable). Understanding where and why the two paths diverge could reveal important nuances in the behavior of early stopping, and we see this as a promising direction for future work.
> > >
> > > **GD vs. regularization for calibration**.
> > >
> > > *Is there a logistic regression example such that early stopped GD has a better calibration/logistic risk rate than $\ell_2$-regularization or vice-versa?*
> > >
> > > This is a great question, as it directly probes the extent to which early stopping replicates the effects of explicit regularization. We currently lack the tools to definitively answer this, but we believe that resolving this would significantly deepen our understanding of early stopping's regularization effect.
> > >
> > > By Theorem 3.1 and discussions in Lines 296–300, both GD and $\ell_2$-regularization require careful tuning—via early stopping or non-vanishing regularization, respectively—to attain good calibration. Although Theorem 5.3 shows that the two paths can diverge asymptotically, our bounds are not sharp enough to yield a clear separation in performance between early stopping and $\ell_2$-regularization in terms of calibration or logistic risk. We will mention this as a concrete open problem in the revision.

---

### Official Review · Reviewer_ViYA · 2025-03-25

**Overall Recommendation:** 4

**Summary:**

The authors examined high-dimensional logistic regression scenarios where p could be finite or infinite. They analyzed the gradient flow dynamics of logistic regression, discussed the generalization capabilities of early stopping estimators and interpolators, and provided comparisons between the gradient descent (GD) path and the L2 penalty path. This topic is particularly interesting for its theoretical insights into optimization and generalization in high-dimensional settings.

**Claims And Evidence:**

I have not checked the proofs carefully, but most of the claims sound reasonable for me.

**Essential References Not Discussed:**

Though the settings are different, I am not sure if the authors ignored the line of work on logistic regression by E. Candes et al. For example: modern maximum-likelihood theory for high-dimensional logistic regression.

**Experimental Designs Or Analyses:**

No experiments have been reported.

**Methods And Evaluation Criteria:**

This is a theoretical paper and no method was proposed.

**Other Comments Or Suggestions:**

N/A

**Other Strengths And Weaknesses:**

The results sound reasonable to me. However, it would be better if the author could provide a more concrete comparison between their work and those from linear regression using square loss. This would help readers better understand the key differences between square loss and logistic loss.

**Questions For Authors:**

Please see the "Other Strengths And Weaknesses".

**Relation To Broader Scientific Literature:**

Most of the current work focuses on regression under square loss. In contrast, regarding logistic regression, most research focuses on empirical minimal loss rather than analyzing gradient descent with early stopping. This work would be of interest to other theoretical groups.

**Theoretical Claims:**

I have not checked the proofs carefully. Most of the claims sound reasonable for me.

---

> ### Author Rebuttal · Authors · 2025-03-31
>
> We appreciate your support! We address your concerns as follows.
>
> ---
>
> **Q1.** “Though the settings are different, I am not sure if the authors ignored the line of work on logistic regression by E. Candes et al.”
>
> **A1.**  We will cite and discuss these works you pointed out in the revision. They focused on the existence of MLE and its behavior if it exists, which is quite apart from our focus where MLE never exists (see paragraph “Noise and overparameterization.” in Section 2 and Proposition 2.2).
>
> ---
>
> **Q2.** “...it would be better if the author could provide a more concrete comparison between their work and those from linear regression using square loss. This would help readers better understand the key differences between square loss and logistic loss.”
>
> **A2.** Risk bounds for logistic regression and linear regression are not directly comparable, as these two problems are different in nature: they have different data assumptions and different risk measurements.
>
> In Lines 244-257 of the right column, we provide a high level comparison on the techniques for analyzing GD in logistic regression and linear regression. One immediate issue is that we cannot directly use tools from linear regression, which relies on the chains of equalities that needs the Hessian to be constant.
>
> We will make these discussions more detailed in the revision to better clarify the key difference between squared loss and logistic loss.

---

> > ### Comment · Reviewer_ViYA · 2025-04-04
> >
> > Thank you for your response. I will keep the current score

---

### Decision · Program_Chairs · 2025-05-01

**Decision:**

Accept (poster)

**Comment:**

This article studies the effect of early stopped gradient descent (GD) on overparametrized logistic regression models, constituting a significant extension to previous works mostly focused on linear regression. Under anisotropic Gaussian feature vectors with bounded covariance trace, and class labels generated from a logistic model with constant misclassification rate, the statistical advantages of early stopped GD are demonstrated in terms of estimation error (measured by logistic loss and calibration error) and classification error. Two upper bounds are established for early stopped GD,  describing first a bias-dominating phase, followed by a variance dominating regime, where a sharp bound is obtained under the stronger condition of bounded parameter vector in norm, These upper bounds implies a polynomial sample complexity for early stopped GD, whereas the derived lower bound for interpolating estimators indicates that asymptotic GD requires at least a exponential number of training samples.

The significance and the novelty of this analysis were appreciated by the reviewers, in addition to other positive points including the clearly structured manuscript, the well-interpreted theoretical results, and the informative discussion of the limitations. The authors adequately answered the reviewers’ questions during rebuttal, resulting in an unanimous recognition of this contribution.